# Transformer-based Planning for Symbolic Regression

**Parshin Shojaee**[* 1] , **Kazem Meidani**[* 2]
**Amir Barati Farimani** [2,3] , **Chandan K. Reddy**[1]
[1] Department of Computer Science, Virginia Tech
[2] Department of Mechanical Engineering, Carnegie Mellon University
[3] Machine Learning Department, Carnegie Mellon University

## Abstract

Symbolic regression (SR) is a challenging task in machine learning that involves finding a mathematical expression for a function based on its values. Recent advancements in SR have demonstrated the effectiveness of pre-trained transformer models in generating equations as sequences, leveraging large-scale pre-training on synthetic datasets and offering notable advantages in terms of inference time over classical Genetic Programming (GP) methods. However, these models primarily rely on supervised pre-training objectives borrowed from text generation and overlook equation discovery goals like accuracy and complexity. To address this, we propose TPSR, a **T**ransformer-based **P**lanning strategy for **S**ymbolic **R**egression that incorporates Monte Carlo Tree Search planning algorithm into the transformer decoding process. Unlike conventional decoding strategies, TPSR enables the integration of non-differentiable equation verification feedback, such as fitting accuracy and complexity, as external sources of knowledge into the transformer equation generation process. Extensive experiments on various datasets show that our approach outperforms state-of-the-art methods, enhancing the model's fitting-complexity trade-off, extrapolation abilities, and robustness to noise [1] .

## 1   Introduction

Symbolic regression (SR) is a powerful method to discover mathematical expressions for governing equations of complex systems and to describe data patterns in an interpretable symbolic form. It finds extensive applications in science and engineering, enabling the modeling of physical phenomena in various domains such as molecular dynamics, fluid dynamics, and cosmology [1–6]. Symbolic representations provide valuable insights into complex systems, facilitating a better understanding, prediction, and control of these systems through the design of accurate, generalizable, and efficient models [7–9]. SR models establish the functional relationship between independent and target variables by mapping them to mathematical equations. The input data can be obtained from simulations, experimental measurements, or real-world observations. Symbolic regression, however, poses several challenges, including the combinatorial nature of the large optimization search space, vulnerability to the quality of input data, and the difficulty of striking a balance between model fitting, complexity, and generalization performance [10, 11].

Symbolic regression encompasses a wide range of methods, spanning different categories. Traditional approaches, such as Genetic Programming (GP), use a heuristic population-based search strategy where each individual represents a potential solution to the problem [12, 13]. Though GP algorithms are capable of finding solutions for nonlinear and complex problems, they are typically slow to converge due to the vast functional search space. Also, as they need to start the search from scratch for each dataset, they tend to be computationally expensive, prone to overfitting, and sensitive to the

---

[*]Equal contribution. Contact email: parshinshojaee@vt.edu
[1]The codes are available at: `https://github.com/deep-symbolic-mathematics/TPSR`

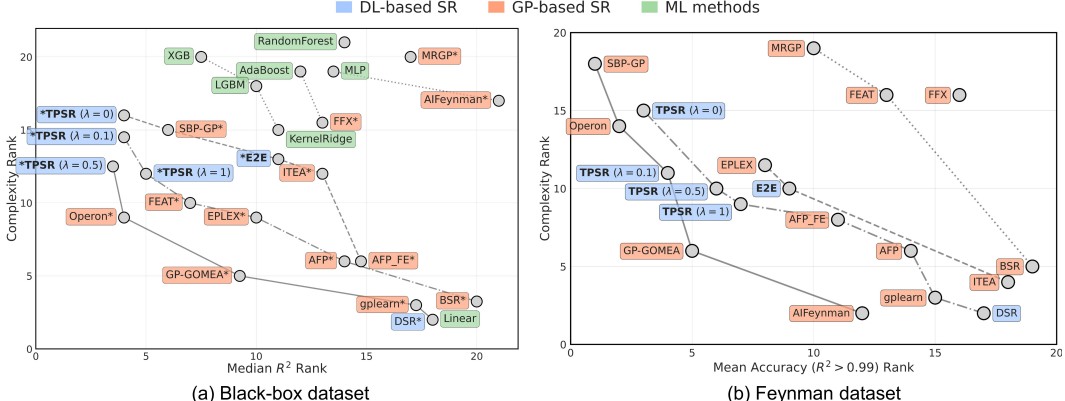

Figure 1: Pareto plot comparing the rankings of all methods in terms of the $R^2$ performance and identified equation complexity for **(a) SRBench *Black-box* datasets** and **(b) *Feynman* datasets**. Our results with Transformer-based Planning (TPSR) applied on top of E2E transformer SR model improves its average accuracy on both data groups while maintaining a similar range of equation complexity. *TPSR can successfully reach the first Pareto-front which is better than E2E baseline on both data groups*. Connecting lines denote Pareto dominance rankings, colors denote the families of models, and "∗" indicates SR versus ML methods in *Black-box* datasets.

choice of parameters [14]. Recent works in SR have shown promising results by using pre-trained transformers [15] for generating equations as sequences of tokens. These models leverage the prior knowledge learned through large-scale pre-training and can generate equations with a single forward pass, leading to considerably faster inference time compared to the GP-based methods [16–19]. However, one of the limitations of these models is that they focus on the supervised language model pre-training objective borrowed from text generation, i.e., they are trained solely with the token-level cross-entropy loss, which can result in equations that may exhibit high token-level similarities but are suboptimal with respect to equation discovery objectives such as fitting accuracy and complexity which are critical in this task. To mitigate this issue, beam search [20, 21] or sampling [22] approaches have been employed as decoding strategies to propose multiple candidate equations for a given dataset, and then select the optimal candidate equation based on the fitting accuracy after optimizing for constants. Nonetheless, both beam search and sampling decoding strategies primarily rely on the pre-trained transformer's logits and next token probabilities, and therefore do not receive any performance feedback during the generation of equation candidates.

To consider the equation discovery objectives in the transformer generation process and still benefit from the pre-trained model logits, we propose TPSR, a **T**ransformer-based **P**lanning strategy for **S**ymbolic **R**egression. TPSR leverages a lookahead planning algorithm, using Monte Carlo Tree Search (MCTS) as a decoding strategy on top of pre-trained transformer SR models to guide equation sequence generation. *TPSR significantly improves performance of the discovered equations by considering verification feedback during the generation process and still remains considerably faster than GP-based models which do not leverage the pre-training priors and learn expressions for each dataset from scratch.* Notably, our approach is model-agnostic and can be applied to any pre-trained SR model, enabling optimization of generated equation sequences for non-differentiable objectives that may encompass combinations of fitting accuracy, complexity, and equation forms. Additionally, we incorporate different caching mechanisms to reduce the overall inference time. Our experimental results demonstrate that applying TPSR on top of the pre-trained E2E SR model [18] significantly enhances its performance across various benchmark datasets. As depicted in Fig. 1, TPSR achieves a strong balance between fitting accuracy and model complexity compared to other leading baselines. It also effectively drives the E2E model towards the optimal trade-off, represented by the first Pareto front. The major contributions of this work are summarized below:

- Proposing TPSR, a new method that combines pre-trained transformer SR models with Monte Carlo Tree Search (MCTS) lookahead planning to optimize the generation of equation sequences while considering non-differentiable performance feedback.

- Developing a new reward function that balances equation fitting accuracy and complexity to optimize the generated equations for an effective trade-off.

- Demonstrating that TPSR consistently outperforms state-of-the-art baselines across various SR benchmark datasets, generating equations with higher fitting accuracy while maintaining lower complexity to avoid non-parsimonious solutions. TPSR still achieves considerably faster inference time than GP-based models which do not use pre-trained priors.

- Showcasing the extrapolation and noise robustness of TPSR compared to the baseline and conducting an ablation study to investigate the impact of various model components.

## 2   Related Work

**Symbolic Regression without Learned Priors.**   Genetic Programming (GP) algorithms are typically employed for single-instance SR, aiming to find the best-fit equation for a "single" dataset at hand [12]. Recently, alternative neural network-based search algorithms have been explored, including deep reinforcement learning (RL) [14, 23, 24], combinations of GP and RL [25], and Monte Carlo Tree Search (MCTS) as a standalone framework [26]. Despite their successes, all these methods lack the benefits of prior knowledge learned from large-scale pre-training. Consequently, they are slow during inference as they need to restart the search from scratch for new datasets.

**Pre-trained Transformers for Symbolic Regression.**   In recent years, pre-trained transformers have shown remarkable performance in natural language and programming language tasks [27–29]. This success has inspired researchers to develop pre-trained transformer models for SR [16–19, 30]. For example, Biggio *et al.* [16] introduced a Neural Symbolic Regression model that scales (NeSymReS) with the amount of synthetic training data and generates equation skeletons where all the numerical constants are represented by a single token "$C$". Kamienny *et al.* [18] proposed an end-to-end transformer SR framework that predicts the complete equation form along with its constants. More recent works [30, 31] introduced unified frameworks that include a transformer-based pre-training stage as the prior for subsequent RL or GP optimization steps. While GP and RL methods have to start anew for each problem, the transformer approaches rely on synthetic data and the power of large-scale pre-trained priors to generate equations in a single forward pass. However, these models are pre-trained on token-level language modeling loss function and thus can perform suboptimal for other equation discovery objectives critical in SR such as fitting accuracy to the observed data as well as equation's complexity. Our model, TPSR, utilizes lookahead planning to guide the generation of equations towards better performance by employing these objectives as feedback during the transformer generation process.

**Planning in Sequence Generation.**   Recently, planning algorithms have been utilized in NLP tasks to optimize text output for specific objectives, such as controlling generated text to meet certain constraints like non-toxicity or conveying certain emotions [32–34]. Recent advances in programming language models developed in code generation have also yielded promising techniques that could be adapted for SR, as they share several vital similarities with each other. Both involve generating sequences of symbols for a given input and typically require optimizing the generated sequences for specific criteria which is different from the pre-trianing objective. For code generation, this may involve optimizing objectives like code compilability, readability, or passing test cases [35–37]. Similarly, in SR, the focus may be on equation-specific sequence-level objectives such as fitting accuracy or minimizing complexity. Motivated by these successes, we develop an approach that combines MCTS planning with pre-trained transformer SR models for improved equation discovery.

## 3   Methodology

### 3.1   Preliminaries

In SR, the main goal is to find a symbolic expression for the unknown function $f(\cdot)$ mapping the $d$-dimensional input $\boldsymbol{x} \in \mathbb{R}^d$ to the target variable $y = f(\boldsymbol{x}) \in \mathbb{R}$. Given a dataset of $n$ observations $\mathcal{D} = (\boldsymbol{x}_i, y_i)_{i=1}^n$, SR methods try to generate an equation $\tilde{f}(\cdot)$ such that $y_i \approx \tilde{f}(\boldsymbol{x}_i)$ for all $i \in \mathbb{N}_n$. Also, the proposed equation is desired to generalize well and to effectively balance the fitting accuracy and complexity. The transformer SR models are trained on a large-scale dataset comprising equation instances paired with their corresponding observations, $\{(\mathcal{D}_1, f_1(\cdot)) \ \ldots \ (\mathcal{D}_M, f_M(\cdot))\}$, where $M$ is the dataset size (number of paired samples). During inference, the trained model directly generates the equation $\tilde{f}(\cdot)$ as a sequence of tokens in an autoregressive manner. An effective way to represent the expression tree of equations in a sequence is to use prefix notation as in [38]. For embedding the

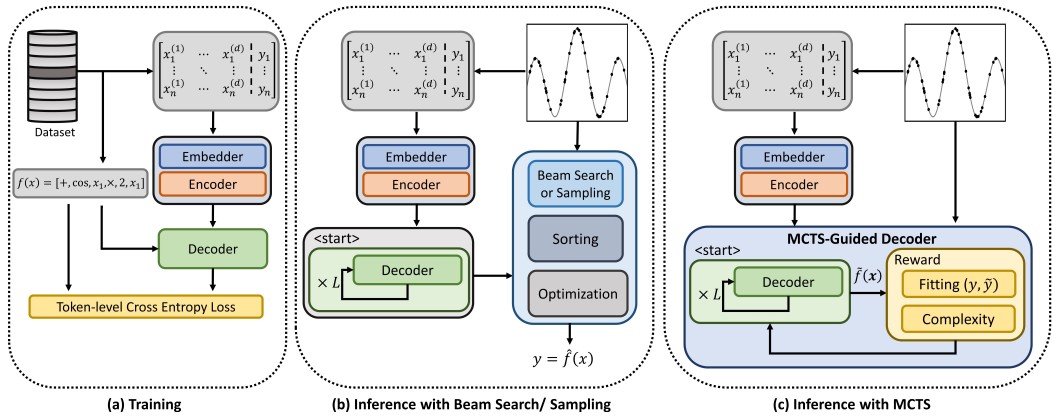

Figure 2: An overview of our proposed method with MCTS-guided decoding at inference compared to the concurrent works with beam search/sampling decoding strategy.

observations, we adopt the pre-trained SR model backbone from [18]. Notably, given the potential for large input sequences with tokenized numeric data and the quadratic complexity of transformers, the method introduced in [18] deploys a linear embedder module to map tokenized inputs to a singular embedding space before introducing them to the transformer encoder and decoder. Subsequent to embedding, these models encode the input observations and then pass the encoded representation along with the masked tokens to decode the equation sequence. To train the model, token-level cross-entropy loss with teacher forcing is employed to learn the distribution of next token prediction conditioned on the encoded dataset and the current state of sequence (Fig. 2(a)).

Achieving a good fitting performance from the model's predicted sequence demands generating accurate constants in the equation. To address this, the generated skeleton or equation can undergo a round of optimization to estimate their constants using nonlinear methods, such as Broyden–Fletcher–Goldfarb–Shanno algorithm (BFGS) [39]. Previous works [18, 16] employ beam search and sampling strategies for transformer decoding in combination with constant optimization to propose several candidate equations. Subsequently, they use fitting metrics such as $R^2$ to order these candidates and output the final equation with the best performance (Fig. 2(b)). Transformer models utilizing beam search or sampling decoding strategies can generate multiple high-likelihood equation sequences, but their generation process is based on logits obtained from model parameters pre-trained with token-matching loss relative to the reference equation. As a result, such models lack the capability to receive verification feedback during generation and optimize sequence for equation discovery objectives such as fitting or complexity of equations.

## 3.2 MCTS-Guided Equation Generation

To generate equations that are both better-fitting and less-complex, it is crucial to incorporate feedback into the equation generation process. To achieve this, we utilize Monte Carlo Tree Search (MCTS) lookahead planning during inference, guiding the decoder towards optimal solutions for fitting and complexity objectives (as shown in Fig. 2(c)). The MCTS-guided transformer decoding explores different possibilities, identifying the most promising paths based on the objectives.

We frame the SR equation generation task as a Markov Decision Process (MDP) where state $s$ represents the current sequence at generation step (token) $t$. If $s$ has not reached the terminal state (i.e., the <EOS> token), we select the next token from the vocabulary as action $a$, updating state $s'$ by concatenating $s$ and $a$. Upon reaching the terminal state, the reward $r$ is computed and used to update the decoding model. MCTS represents states as nodes and actions as edges within a tree structure, navigating state-space from the root node (i.e., initial state) to reach terminal states with maximum rewards. MCTS balances exploration and exploitation, considering nodes that lead to higher quality equations (i.e., higher Q-values) and under-explored nodes (i.e., those with fewer visits). During the generation process of the transformer, we utilize the MCTS algorithm iteratively to conduct lookahead planning and determine the next token. However, the large search-space requires more than the sole application of MCTS to discover high-quality equations. We need to effectively share information between the pre-trained transformer model and MCTS for better generations. To achieve this, we incorporate the probabilities of the next-token that are acquired through the pre-trained

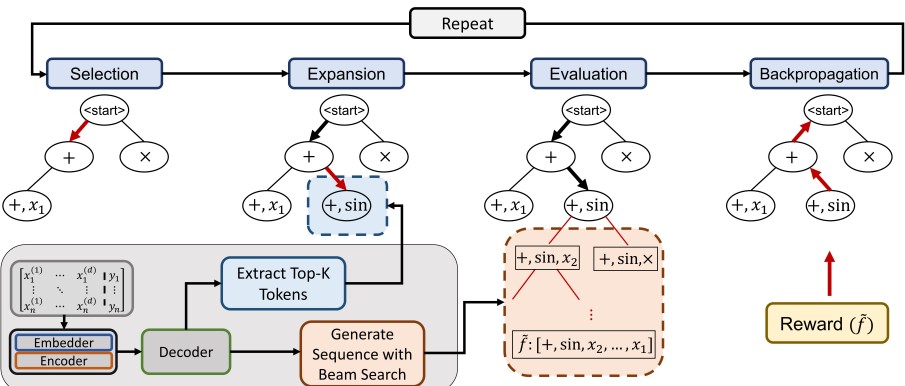

Figure 3: Overview of TPSR's key steps: Selection, Expansion, Evaluation, and Backpropagation. MCTS-guided lookahead planning in decoding interacts with the pre-trained transformer SR model in the expansion and evaluation steps employing the transformer $top\text{-}k$ sampling and beam search, respectively. The designed reward is used to guide the backpropagation.

transformer SR models into the MCTS planning process. This incorporation helps to enhance the search process, leading to more efficient and effective results. The key steps of MCTS for transformer decoding in SR models, as depicted in Fig. 3, are as follows:

**Selection.** The Upper Confidence Bound for Trees [40] criterion is employed to select actions (i.e., next tokens) for fully extended nodes in the search tree, balancing exploration and exploitation. We use the P-UCB heuristic [41] as

$$\text{P-UCB}(s,a) = Q(s,a) + \beta \cdot P_\theta(a|s) \cdot \sqrt{\frac{\ln\left(N(s)\right)}{1+N(s')}}, \tag{1}$$

where $Q(s,a)$ is the maximum return for action $a$ in state $s$ across all simulations, promoting the exploitation of the optimal child node. The second term encourages exploration of less-visited children, with $N(s)$ as state $s$'s visit count and $s'$ as the subsequent state. $P_\theta(a|s)$ is the probability of the next token $a$ given the partial sequence state $s$ from pre-trained transformer model parameterized by $\theta$. The exploration-exploitation trade-off is adjusted by hyperparameter $\beta$. Lastly, the next token action maximizes the P-UCB: $\text{Select}(s) = \arg\max_a \text{P-UCB}(s,a)$.

**Expansion.** In the expansion stage, after selecting a node that is not fully expanded, a new child (next token) for the current state is explored. Random expansion of the node from the vocabulary, however, might result in an invalid equation (that does not comply with the prefix notation) and makes the search process very time-consuming. Therefore, given partial equations, only $top\text{-}k$ most likely choices of the next token are considered as the possible children of the node for expansion. In other words, we are restricting the actions to be only from the $top\text{-}k$ high-likelihood options which are retrieved from the pre-trained transformer SR model's logits. These options are then ordered to determine the sequence in which the children will be expanded.

**Evaluation.** To evaluate the newly expanded nodes, we perform simulations to complete the equation sequence. This is necessary because the new state may still be a partial equation and performance feedback can only be obtained at the end of the sequence when the equation generation is completed. In MCTS, it is common to employ random actions during the simulation stage. Nevertheless, random action selection for equation generation, much like during expansion, suffers from certain drawbacks in terms of time and the possibility of generating invalid equations. Consequently, the pre-trained transformer SR model is invoked again, this time utilizing beam search with a beam size of $b$, to generate complete equation candidates based on the current state. The beam size $b$ determines the number of complete equations to be generated from the current partial equation. Following these simulations, the highest reward among all the complete equation candidates is assigned to the new node value.

**Backpropagation.** After generating a complete equation $\tilde{f}(\cdot)$, the corresponding reward $r(\tilde{f}(\cdot))$ can be computed. The highest reward among all simulations is then assigned to the new node, which recursively backpropagates its estimated value to its parents until it reaches the root of the tree. This

update process involves updating the $Q$ values of all state-action pairs, denoted as $s'$ and $a'$, along the trajectory in the tree to reach the root. Specifically, for each state-action pair, the $Q$ value is updated by taking the maximum of the current $Q$ value and the new value $r$: $Q(s', a') \leftarrow \max(Q(s', a'), r)$.

More details on TPSR, including its steps and implementation can be found in Appendix C.

### 3.3 Reward Definition

We define a numerical reward $r \in \mathbb{R}$ to evaluate complete equation candidate $\tilde{f}(\cdot)$, promoting fitting accuracy and regulating complexity. After optimizing constants in the complete sequence, we compute the reward. We first calculate the normalized mean squared error (NMSE) between ground-truth target variable $y$ and predicted target variable $\tilde{y} = \tilde{f}(\boldsymbol{x})$, and formulate the reward as:

$$r(\tilde{f}(\cdot)|\boldsymbol{x}, y) = \frac{1}{1 + \mathrm{NMSE}(y, \tilde{f}(\boldsymbol{x}))} + \lambda \exp\left(-\frac{l(\tilde{f}(\cdot))}{L}\right), \tag{2}$$

where $l$ represents equation complexity as the sequence length in prefix notation [18, 42, 16]; $L$ denotes the model's maximum sequence length; and $\lambda$ is a hyperparameter balancing fitting and complexity reward terms. Higher $\lambda$ values favor less complex equations, encouraging best-fitting and penalizing non-parsimonious solutions. NMSE is calculated as $\left(\frac{1}{n}\left\|y - \tilde{f}(\boldsymbol{x})\right\|_2^2\right)/\left(\frac{1}{n}\left\|y\right\|_2^2 + \epsilon\right)$, where $\epsilon$ is a small constant to prevent numerical instability.

### 3.4 Efficient Implementation with Caching

During MCTS evaluation, the transformer model generates complete equation sequences from a given state, constructing implicit tree structures for beam search and computing $top\text{-}k$ next tokens for visited states. These computations are required in future MCTS iterations, so we employ two caching mechanisms, $top\text{-}k$ *caching* and *sequence caching*, to reduce redundancy and improve efficiency. $Top\text{-}k$ *caching* stores computed $top\text{-}k$ values for given states. For example, in Fig. 4, when evaluating state $s = [+, \sin]$ in MCTS iteration $t$, $top\text{-}k$ tokens are computed for $s$ and subsequent visited states, such as $[+, \sin, x_2]$. State and $top\text{-}k$ value pairs are cached for future use, avoiding re-

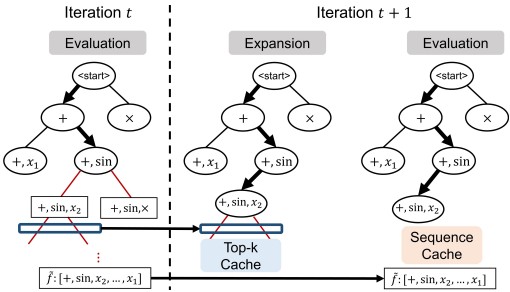

Figure 4: An illustration of caching mechanisms in TPSR.

dundant token retrieval. *Sequence caching* caches complete equations generated with the provided beam size. If a state matches a stored equation partially, the cached equation can be used directly in future iterations, bypassing iterative sequence generation. Both caching strategies are designed to enhance efficiency without compromising performance. More details are provided in Appendix C.

## 4 Experiments

In this section, we present our experimental results that evaluate the effectiveness and efficiency of TPSR. While the proposed decoding strategy is generally model-agnostic, here we showcase the results of using TPSR for the end-to-end (E2E) pre-trained SR transformer backbone [18], as E2E is the state-of-the-art open-source pre-trained SR model with publicly accessible model weights. Additional results of using TPSR with the NeSymReS pre-trained SR backbone [16] can be found in Appendix D.4. We evaluate our framework by answering the following research questions (**RQs**):

**RQ1.** Does TPSR perform better than other decoding strategies (beam search/sampling) and competing baseline methods over standard SR benchmark datasets?

**RQ2.** Does TPSR provide better extrapolation and robustness to noise?

**RQ3.** Are TPSR's caching mechanisms effective in reducing computation time?

**RQ4.** What is the role of individual components in TPSR's overall performance gain?

Table 1: Performance of TPSR compared with beam search and sampling decoding strategies on the SRBench [42] and In-domain Synthetic [18] datasets.

| Data Group | Model | Feynman | | Strogatz | | Black-box | |
|---|---|---|---|---|---|---|---|
| | | $\uparrow R^2 > 0.99$ | $\downarrow$ Complexity | $\uparrow R^2 > 0.99$ | $\downarrow$ Complexity | $\uparrow R^2$ | $\downarrow$ Complexity |
| | E2E+Beam | 0.815 | 54.19 | 0.357 | 53.21 | 0.847 | 83.61 |
| | E2E+Sampling | 0.848 | 50.73 | 0.357 | 50.14 | 0.864 | 82.78 |
| SRBench | TPSR ($\lambda$=0) | **0.952** | 84.42 | **0.928** | 82.78 | 0.938 | 129.85 |
| | TPSR ($\lambda$=0.1) | 0.949 | 57.22 | 0.785 | 56.14 | **0.945** | 95.71 |
| | TPSR ($\lambda$=0.5) | 0.924 | 50.01 | 0.714 | 47.02 | 0.931 | 82.58 |
| | TPSR ($\lambda$=1) | 0.916 | **47.24** | 0.571 | **43.42** | 0.924 | **79.43** |

| Data Group | Model | $\uparrow R^2 > 0.99$ | $\uparrow R^2$ | $\uparrow Acc_{0.1}$ | $\uparrow Acc_{0.01}$ | $\uparrow Acc_{0.001}$ | $\downarrow$ Complexity |
|---|---|---|---|---|---|---|---|
| | E2E+Beam | 0.657 | 0.782 | 0.461 | 0.298 | 0.2 | 38.37 |
| | E2E+Sampling | 0.640 | 0.794 | 0.472 | 0.332 | 0.208 | 39.82 |
| In-domain | TPSR ($\lambda$=0) | 0.702 | 0.828 | **0.550** | **0.416** | **0.333** | 67.11 |
| | TPSR ($\lambda$=0.1) | **0.708** | **0.833** | 0.514 | 0.326 | 0.213 | 40.31 |
| | TPSR ($\lambda$=0.5) | 0.697 | 0.830 | 0.459 | 0.274 | 0.184 | 36.55 |
| | TPSR ($\lambda$=1) | 0.691 | 0.827 | 0.439 | 0.271 | 0.176 | **35.67** |

## 4.1 Datasets

We evaluate TPSR and various baseline methods on standard SR benchmark datasets from Penn Machine Learning Benchmark (PMLB) [43] studied in SRBench [42], as well as *In-domain Synthetic Data* generated based on [38, 18]. The benchmark datasets include 119 equations from *Feynman Lectures on Physics database* series[2] [44], 14 symbolic regression problems from the *ODE-Strogatz database*[3] [45], and 57 *Black-box*[4] regression problems without known underlying equations. We limit the datasets to those with continuous features and input dimension $d \leq 10$, as the transformer SR model [18] is pre-trained with $d_{max} = 10$. The *In-domain Synthetic Data* consists of 400 validation functions with different levels of difficulty and number of input points. This data is referred to as "in-domain" because the validation functions and their corresponding input points are generated using the same approach as the data on which the backbone transformer model [18] is pre-trained. More details on each of these datasets are provided in Appendix A.

## 4.2 Evaluation Metrics

We evaluate our model using the following three metrics: $R^2$ score [42], accuracy to tolerance $\omega$ [16, 46], and equation complexity [18, 42].

$$R^2 = 1 - \frac{\sum_i^{N_{test}}(y_i - \tilde{y}_i)^2}{\sum_i^{N_{test}}(y_i - \bar{y})^2}, \quad Acc_\omega = \mathbb{1}\big(\max_{1 \leq i \leq N_{test}} \left|\frac{\tilde{y}_i - y_i}{y_i}\right| \leq \omega\big), \quad Complexity = \left|\mathcal{T}(\tilde{f}(\cdot))\right|,$$

where $R^2$ measures fitting performance with $\bar{y}$ as the mean of $y$ in test set, $Acc_\omega$ evaluates equation precision based on tolerance threshold $\omega$, and equation complexity is determined by the number of nodes in the expression tree $\mathcal{T}$ of the generated equation $\tilde{f}(\cdot)$. Following [18, 42], we set $R^2 = 0$ for rare pathological examples and discard the worst 5% predictions for $Acc_\omega$ to reduce outlier sensitivity.

## 4.3 (RQ1) Effectiveness of TPSR

Table 1 presents the performance comparison results of TPSR with the baseline decoding strategies on the SRBench benchmark and the In-domain synthetic dataset. For the E2E baseline, we use the settings reported in [18], including beam/sample size of $C = 10$ candidates, and the refinement of all the candidates $K = 10$. For our model, we use the width of tree search as $k_{max} = 3$, number of rollouts $r = 3$, and simulation beam size $b = 1$ as the default setting. For PMLB datasets that contain more than 200 points, we follow [18] and use $B$ bags of data, each containing $N = 200$ points, due to the limitation that the baseline method is pre-trained with $N \leq 200$ data points. In the baseline method [18], a total of $BC$ candidates are generated ($C$ candidates for $B$ bags), which are then sorted and refined to generate the best equation. However, for TPSR, since we need to train an MCTS for each bag, we use an iterative decoding approach, starting with the first bag and continuing

---

[2]`https://space.mit.edu/home/tegmark/aifeynman.html`
[3]`https://github.com/lacava/ode-strogatz`
[4]`https://github.com/EpistasisLab/pmlb/tree/master/datasets`

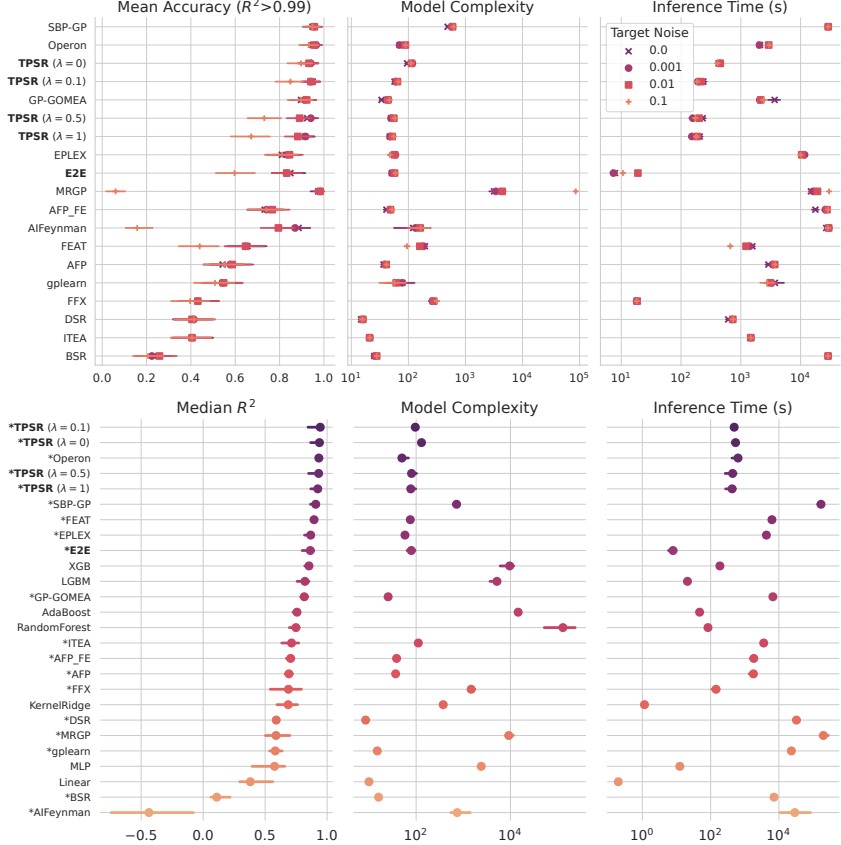

Figure 5: Performance comparison of TPSR and SRBench algorithms in terms of Accuracy-Complexity-Time on *Feynman* (top) and *Black-box* (bottom) datasets. For *Feynman* dataset, algorithms are sorted based on mean accuracy defined as the ratio of solutions with $R^2 > 0.99$ on test set under various noise levels, and for *Black-box* dataset, the algorithms are sorted based on the median $R^2$ score on test set. TPSR demonstrates a strong balance of performance with relatively low model complexity and lower inference time compared to GP-based algorithms. The error bars represent the 95% confidence interval and "∗" refers to SR methods for *Black-box* dataset.

with subsequent bags until a criterion ($R^2 > 0.99$) is met or we use a maximum of $B = 10$ bags. To ensure a fair comparison, we use $B = 10$ for the E2E baseline method as well. In this table, we demonstrate the results of our proposed framework, TPSR, with varying values of the $\lambda$ parameter that controls the trade-off between fitting performance and complexity in the hybrid reward function defined in Eq. (2). For a detailed comparison of the experimental settings across different approaches, refer to Table 2 in Appendix B.

As shown in Table 1, when $\lambda = 0$, the framework generates complex equations that overoptimize for fitting performance. However, as we increase $\lambda$, the framework generates less complex equations with a slight reduction in fitting performance. Notably, even for large values of $\lambda$, such as $\lambda = 1$, the fitting performance of TPSR significantly outperforms that of the baseline methods. Based on the results, we recommend a default setting of $\lambda = 0.1$ as it offers a balanced trade-off between complexity and accuracy, while also mitigating potential overfitting (as detailed in Appendix D.1). These findings demonstrate the superiority of TPSR over the baseline methods in terms of fitting performance across all datasets, while generating equations with comparable or reduced complexity than those generated by the baseline methods. Table 1 shows that TPSR exhibits a more significant gap in fitting performance when compared to E2E baselines on SRBench datasets, while this gap is smaller for In-domain datasets (even performing slightly worse on $Acc_\omega$ for larger $\lambda = 0.5, 1$). This is due to the In-domain dataset being generated using the same approach as the E2E pre-training data, resulting in the E2E model's superior performance on this synthetic dataset. Furthermore, qualitative comparisons of TPSR with baseline symbolic and black-box regression models demonstrate the

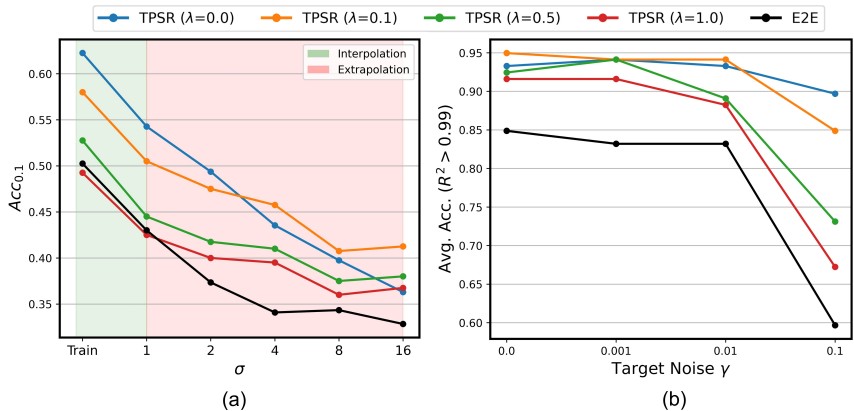

Figure 6: TPSR with $\lambda \in \{0, 0.1, 0.5, 1\}$ compared to E2E for **(a) Extrapolation performance** where in-domain accuracy is shown for different input sampling variance ($\sigma$), and **(b) Robustness to noise**, where mean accuracy ($R^2 > 0.99$) is shown for various target noise levels ($\gamma$).

superior performance of TPSR in learning the underlying equation and out-of-domain extrapolation (see Appendix D.3).

Fig. 5 presents a detailed comparison of our proposed TPSR with the baseline E2E transformer model and all the SRBench baselines on the PMLB *Feynman* and *Black-box* datasets. This figure illustrates the relative position of each algorithm with respect to (1) fitting performance, (2) model complexity, and (3) inference time. The results indicate that transformer-based planning in the TPSR significantly enhances the performance of E2E and outperforms even the state-of-the-art GP baselines, achieving the highest fitting performance on the black-box datasets. This is achieved while the complexity of the generated equations in TPSR is not greater than that of E2E, and shows a great fitting-complexity-time balance compared to other SR algorithms. The pareto plots provided in Fig. 1 and Appendix D.2 also demonstrate the effectiveness of TPSR in balancing fitting-complexity as well as fitting-time compared to all other SRBench baselines. Our TPSR effectively pushes this balanced performance to the first pareto front for both the *Feynman* and *Black-box* datasets. Moreover, it is important to note that, while the inference time of TPSR is longer than the baseline E2E transformer model, it still has significantly lower inference time than RL or GP-based SRBench baselines. Further results on the SRBench and In-domain datasets are provided in Appendix D.

### 4.4 (RQ2) Extrapolation and Robustness

The ability to extrapolate well is inherently linked to the quality of the equations discovered through symbolic regression. To investigate the extrapolation performance of TPSR to out-of-training regions, we normalize the input test data points to different scales ($\sigma$) instead of unit variance (used for training points) as per [18]. Fig. 6(a) depicts the average performance of TPSR compared to E2E with sampling decoding on the training data as well as testing data in scales of $\sigma \in \{1, 2, 4, 8, 16\}$ for the *In-domain Synthetic* dataset. Also, we investigate the effect of different complexity controlling levels ($\lambda \in \{0, 0.1, 0.5, 1.0\}$) on the extrapolation performance. It can be observed that, while $\lambda = 0$ (i.e., no complexity regularization) achieves the best fitting accuracy on the training data, it has a sub-par performance for $\sigma > 8$. This can be due to the overfitting issue when the symbolic model is much more complex than the true function, similar to the common overfitting issue in ML models. Results highlight the importance of controlling complexity in the extrapolation of identified equations. For values of $\lambda > 0$, the overfitting issue is mitigated as the generated equations become less complex. However, very high values of $\lambda$ (e.g., $\lambda = 1$) mostly result in poor accuracy performance. The flexibility of TPSR for allowing different values of $\lambda$ to balance fitting and complexity for a given task is crucial for optimal performance. Fig. 6(b) also presents the robustness of TPSR with different $\lambda$ levels compared to the E2E transformer baseline on the *Feynman* dataset. The results indicate that MCTS-guided lookahead planning can offer robust performance with a smaller drop in accuracy compared to the baseline in the presence of noise.

### 4.5 Ablation Study

In this section, we investigate the effect of different MCTS parameters and caching mechanisms on the performance of TPSR by conducting ablation experiments on the *Feynman* datasets.

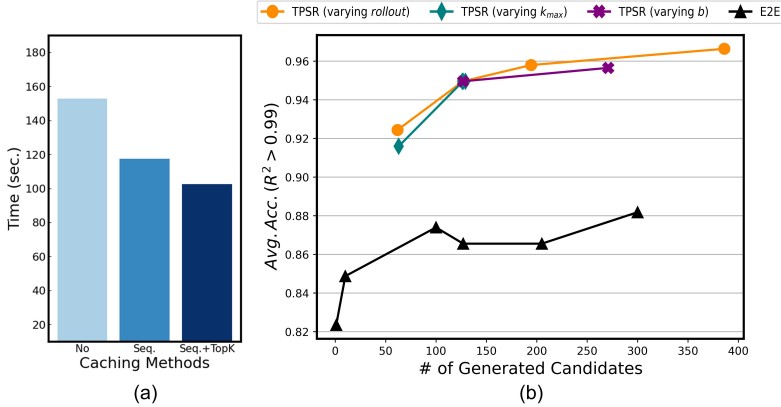

Figure 7: Ablation study on the modules and parameters of TPSR. **(a) Effect of caching mechanisms:** *Sequence caching* and $top\text{-}k$ *caching* improve the inference time of TPSR ($\lambda = 0.1$). **(b) Efficiency and parameters of TPSR**: Average accuracy of TPSR (varying model parameters), and baseline E2E (varying sampling size) across number of generated candidates.

**(RQ3) Caching Mechanisms.** In Fig. 7(a), we illustrate the effectiveness of the *sequence* and $top\text{-}k$ caching mechanisms in reducing the total inference time of TPSR ($\lambda = 0.1$). Our experiments show that sequence caching has more effect in dropping the inference time as it replaces the time-consuming sequence generation process. Overall, these two mechanisms can reduce the total inference time by around $28\%$.

**(RQ4) Search Parameters.** Fig. 7(b) shows the accuracy performance against the number of generated equations during the decoding process for both TPSR ($\lambda = 0.1$) and the baseline E2E with sampling decoding. In this figure, the 'number of generated equation candidates' represents the total number of complete equation sequences generated by each method. Specifically, this refers to the sample size in the E2E with sampling decoding, and the function calls of the beam search sub-routine multiplied by beam size $b$ in TPSR. The results show that under the same number of generated equation candidates, TPSR significantly outperforms the E2E baseline. This is primarily attributed to the fact that the E2E baseline is deprived of any performance feedback during the equation generation process. We report the results for variants of TPSR with different MCTS parameters. We assess the performance with varying number of rollouts, $r = \{1, 3, 6, 9\}$, number of beams in simulations, $b = \{1, 3\}$, and the maximum number of possible expansions at each state, $k_{max} = \{2, 3, 4\}$. The default setting of TPSR parameters are $b = 1$, $k_{max} = 3$, and $r = 3$. Results indicate that increasing $r$, $k_{max}$, and $b$ all contribute to the better performance of TPSR, with the most significant improvement observed when increasing $r$. This is because more rollouts provide model with more opportunities to learn from trials and learn better values.

## 5   Conclusion

In this work, we propose TPSR, a model-agnostic decoding strategy for symbolic regression that leverages the power of pre-trained SR transformer models combined with MCTS lookahead planning, and outperforms the existing methods in generating equations with superior fitting-complexity-time trade-off. We demonstrate the flexibility of TPSR in controlling discovered equation complexity without fine-tuning the pre-trained model. We also show that TPSR performs 100x faster than state-of-the-art genetic algorithms by leveraging the pre-trained priors. Additional results show that better expressions obtained with lookahead planning can further improve model performance in terms of noise robustness and extrapolation to unseen data. Future research could focus on enhancing the adaptability of feedback-based expression generation mechanisms, potentially by modulating the flexibility of MCTS or transformer model weights, and the integration of MCTS with the training or fine-tuning of transformer SR models. Furthermore, employing parallelization and distributed computing could potentially improve planning efficiency.

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
