# Appendix

## A  Dataset Details

We evaluate TPSR and several baseline methods on the following four standard benchmark datasets: *Feynman*, *Black-box*, and *Strogatz* from SRBench [42], and *In-domain Synthetic Data* generated based on [18]. More details on each of these datasets are given below.

***Feynman***[5]**:**  This dataset contains 119 equations sourced from *Feynman Lectures on Physics database* series [44]. The regression input points $(x, y)$ from these equations are provided in Penn Machine Learning Benchmark (PMLB) [42, 43] and have been studied in SRBench [42] for the symbolic regression task. The input dimension is limited to $d \leq 10$ and the true underlying function of points is known. We split the dataset into $B$ bags of 200 input points (when $N$ is larger than 200) since the transformer SR model is pretrained on $N \leq 200$ input points as per [18].

***Strogatz***[6]**:**  This dataset comprises 14 symbolic regression problems sourced from the *ODE-Strogatz database* [45] for nonlinear dynamical systems. The input points for these problems are included in PMLB [43] and have been examined in SRBench [42] for symbolic regression. The input dimension for these problems is restricted to $d = 2$ and the true underlying functions are provided.

***Black-box***[7]**:**  The black-box regression datasets from PMLB [43] are used for the symbolic regression task and studied in SRBench [42] among various baselines. The aim of SR study on these black-box datasets is to find an interpretable model expression that fits the data effectively. We limit the datasets to those with continuous features and input dimension $d \leq 10$, as the transformer SR model [18] is pretrained with $d_{max} = 10$. In total, there are 57 black-box datasets that consist of real-world and synthetic datasets with varying levels of noise.

***In-domain Synthetic Data***:  Following [18], we construct a fixed validation set consisting of 400 equation examples in which the validation functions were uniformly distributed across three different difficulty factors: input dimension ($d$), number of unary operators ($u$), and binary operators ($b$). Specifically, we set $d \sim \mathcal{U}(1, d_{max})$, $b \in \mathcal{U}(d-1, d+b_{max})$, and $u \sim \mathcal{U}(0, u_{max})$, where $d_{max} = 10$, $u_{max} = 5$, and $b_{max} = 5 + d$. The equation sequence is generated for each function by providing $N = [50, 100, 150, 200]$ input points $(x, y)$, and the prediction accuracy is assessed on $N_{test} = 200$ points that are randomly extracted from a multi-center distribution, as described in [18]. This data is referred to as "in-domain" because the validation data is generated using the same approach as the data on which the model [18] is pre-trained.

## B  Implementation Details

Our model implementation leverages the state-of-the-art open-source End-to-End (E2E) SR model [18] as the pre-trained transformer backbone. This selection is due to the public availability of E2E's model architecture, weights, and logits in the Facebook `symbolicregression` library [8] and repository [9]. The algorithm of our model is provided in Appendix C and all the implementation code for our experiments with configuration details for reproducibility are open-sourced: `https://github.com/deep-symbolic-mathematics/TPSR`. In our experiments, the model's maximum sequence length is set to $L = 200$, and the constant to prevent numerical stability $\epsilon$ in NMSE calculation $(\frac{1}{n}\|y - \tilde{f}(x)\|_2^2)/(\frac{1}{n}\|y\|_2^2 + \epsilon)$ is set to $1e-9$. We set the default maximum number of node expansions ($k_{max}$) to be 3, the beam size of simulations ($b$) as 1, and the number of rollouts ($r$) as 3. The complexity-controlling parameter ($\lambda$) was also varied across four values: $0, 0.1, 0.5, 1$. To ensure consistency with the protocol set out by [18], we divided the observation points of each equation in the SRBench datasets (including *Feynman*, *Strogatz*, and *Black-box*) into training and testing sets at a ratio of $75\%/25\%$. In the evaluation experiments involving *In-domain Synthetic Data*, we adjusted the number of observation points for each equation on which TPSR was trained to $N \in [50, 100, 150, 200]$. The generated expression was subsequently tested on the $N_{test} = 200$ data points for each sampling variance ($\sigma$) of 1, 2, 4, 8, and 16. These synthetic input points with varying sampling variance are introduced in *In-domain* data [18] to assess the models' extrapolation

---

[5]`https://space.mit.edu/home/tegmark/aifeynman.html`
[6]`https://github.com/lacava/ode-strogatz`
[7]`https://github.com/EpistasisLab/pmlb/tree/master/datasets`
[8]`https://dl.fbaipublicfiles.com/symbolicregression/`
[9]`https://github.com/facebookresearch/symbolicregression`

Table 2: Experimental Settings of TPSR and E2E [18]

| Setting/Parameter | TPSR | E2E |
|---|---|---|
| Maximum Equation Length ($L$) | 200 | 200 |
| Maximum No. of Observations ($N$) | 200 | 200 |
| Maximum Input Dimension ($d_{max}$) | 10 | 10 |
| Maximum No. of Bags ($B$) | 10 | 10 |
| Beam/Sample size ($C$) | – | 10 |
| No. of Refinement Candidates ($K$) | – | 10 |
| Maximum Expansion Width ($k_{max}$) | 3 | – |
| Maximum No. of Rollouts ($r$) | 3 | – |
| Beam Size in Simulations ($b$) | 1 | – |
| UCT Exploration Parameter ($\beta$) | 1 | – |

capabilities under different conditions. All experiments are implemented with PyTorch on four Quadro RTX 8000 GPUs, with 48GB of RAM.

# C  Methodology Details

## C.1  MCTS-Guided Decoding Details

Algorithm 1 provides the details of steps in MCTS-guided lookahead planning as a decoding strategy for SR. Here, the blue lines correspond to the utilization of reward and selection functions defined in Eqs. (2) and (1) of Section 3. These functions play a crucial role in guiding the MCTS-based Transformer Decoding strategy for SR and ensuring effective exploration and exploitation within the search space. Meanwhile, the red lines in the algorithm denote the places when the pre-trained transformer SR model is invoked to extract the *top-k* next tokens and equation candidate beams. These extracted tokens and beams are employed in the expansion and evaluation steps of the MCTS algorithm, respectively. By incorporating the pre-trained transformer SR model, the MCTS-guided decoding strategy can effectively leverage the model's inherent semantic knowledge gained through large-scale pre-training to generate high-quality equation candidates and enhance the overall performance of the SR approach. Notably, in this MCTS setting, a "visit" signifies that a state-action pairing $(s, a)$, has been explored during tree search, appending the corresponding child state, $s'$, to the tree. Sequences that are generated as part of the beam search sub-routine of simulations in the evaluation stage of MCTS are not directly considered as visits to the nodes corresponding to these sequences. Instead, they serve the purpose of completing the partial equation to allow for feedback computation. As for cache hits, they are also not counted as visits. The reason is that caching in this context is used to save computation by storing previously computed values, and a cache hit simply means retrieving a stored value rather than performing a new visit.

---

**Algorithm 1:** MCTS-Guided Decoding for Symbolic Regression

---

**Input** : $r_{max}$: maximum number of rollouts, $k_{max}$: number of children of nodes used for *top-k* next token selection, $b$: beam size, $c$: P-UCB exploration parameter

**while** $r < r_{max}$ **do**

    $node \leftarrow root$;

    1) Selection

    **while** $|node.children| > 0$ **do**

        node $\leftarrow$ SELECT($node.children$, $c$);

    **end**

    2) Expansion

    next tokens $\leftarrow$ TOP_K($node$, $k_{max}$);

    **for** $action \in next\ tokens$ **do**

        next state $\leftarrow$ CONCAT($node$, $action$);

        Add next state as a child of $node$;

    **end**

    3) Evaluation

    Equation $\leftarrow$ BEAM_SEARCH($node$, $b$);

    $reward \leftarrow$ GET_REWARD(Equation);

    Save (Equation, $reward$) pair in a dictionary ;

    4) Backpropagation

    Update values on the trajectory given the $reward$ ;

**end**

Return Equation with the highest $reward$ ;

---

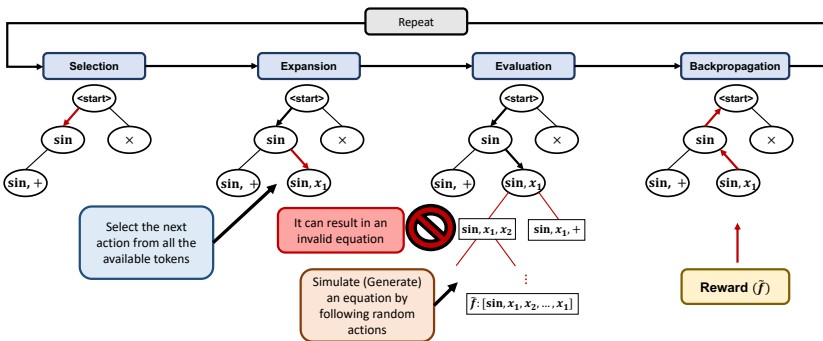

Figure 8: MCTS-Guided decoding algorithm for Symbolic Regression without using the pretrained transformer SR model for expansion and evaluation steps.

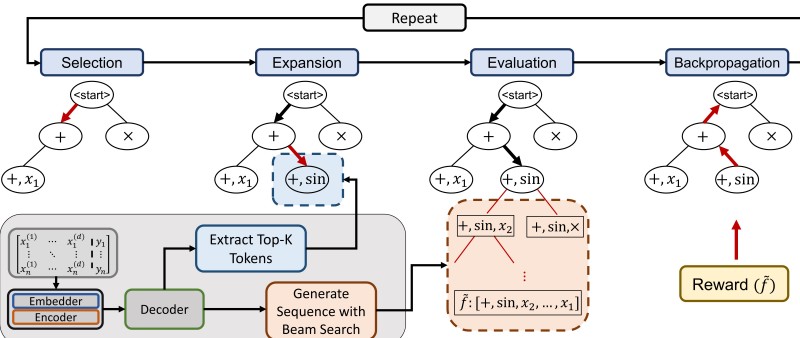

Figure 9: MCTS-Guided decoding algorithm for Symbolic Regression with the pre-trained transformer model used for expansion and evaluation steps.

## C.2 Distinguishing TPSR from other MCTS Approaches in SR

It is essential to highlight that the implementation of the MCTS approach in TPSR differs from the standalone MCTS algorithm for SR. In a recent work, Sun *et al.* [26] shows that Monte Carlo Tree Search can be effective for exploring the optimal expression trees that govern nonlinear dynamical systems. This work introduces several adjustments to the conventional MCTS to enable the recovery of equations as expression trees. However, we would like to remark that using MCTS as a standalone algorithm for SR is a single-instance SR method, meaning that it requires searching from scratch for a new function or measurement data, and does not leverage pre-trained priors. To highlight the role of pre-trained transformer in our TPSR framework, we compare the MCTS-guided decoding algorithm in TPSR (Fig. 9, replicated from the main body for ease of comparison) with a standard MCTS algorithm (Fig. 8) which can be used in a similar fashion but without sharing information with the pre-trained transformer. During the expansion phase, standard MCTS chooses the next accessible action from the action set (i.e., the vocabulary of tokens) and appends the state that can be reached through the chosen action. In this example, action $x_1$ is selected, and the new state appended to the tree is $[sin, x_1]$. Subsequently, during the evaluation phase, MCTS assesses the new state by implementing a *random* policy from the new state and calculating the policy's value. Applying the standard MCTS algorithm to domains characterized by extensive state or action spaces, such as symbolic regression with a large combinatorial optimization space that exponentially grows with the number of input variables, is highly impractical. This is because attempting all possible actions in the expansion phase is infeasible. Furthermore, the random policy employed in the evaluation phase exhibits significant variance when estimating the new state's value, and may result in invalid equations that are unsuitable for proper evaluation (e.g., accurately assessing the equation's fitting performance). To overcome these limitations, TPSR employs the pre-trained transformer SR model. This approach leverages the semantic knowledge embedded in large-scale pre-trained priors, while conducting lookahead planning to optimize equation generation for the equation discovery non-differentiable objectives. By integrating the pre-trained transformer SR model, TPSR can efficiently and effectively

navigate the vast search space, reducing complexity and enhancing fitting performance, thus offering more viable solutions for this task.

It is also crucial to emphasize how the integration of MCTS in TPSR differentiates from others, particularly from works like Kamienny *et al.* [47], which also pairs MCTS with pre-trained transformers. Key differentiators include:

**General Approach.** Unlike [47] that exploits a pre-trained mutation policy $M$ to generate the expression by following a series of mutations from an empty expression (root), TPSR follows the seq2seq approach of E2E [18] to generate the expression token-by-token. Consequently, TPSR uses the pre-trained E2E as its backbone but [47] pre-trains the mutation policy from scratch.

**MCTS and Search Strategy.** In [47], the search tree consists of full mathematical equations, with each node representing a distinct equation and edges corresponding to mutations between equations. In contrast, TPSR employs MCTS as a decoding strategy in the context of the transformer model. Each node in the search tree of TPSR represents the current state of generated tokens, potentially forming non-complete sequences, with edges corresponding to mathematical operators or variables. So, the search tree of [47] with "n" nodes includes "n" different equations, while the TPSR search tree includes partial decoded sequences, and completed equations only exist at the terminal leaf nodes. This distinction inherently leads to major differences in selection, expansion, and back-propagation mechanisms within the MCTS algorithm.

**Parameter Update and Learning.** [47] utilizes MCTS to update and learn the distribution of mutations for a group of out-of-distribution datasets. The approach involves fine-tuning an actor-critic-like model to adjust the pre-trained model on a group of symbolic regression instances. On the other hand, TPSR uses the pre-trained transformer's weights to guide the expansion during the search process, without updating any specific parameters for in-domain or out-of-domain datasets (without fine-tuning). Consequently, the same settings and pre-trained model are applied to both in-domain and out-of-domain evaluations in TPSR.

**Computation Time.** [47] involves pre-training a mutation policy, a critic network, and performing fine-tuning stages for these networks, leading to significantly higher computation time (a limit of 24hrs and 500K equation candidate evaluations as stated in [47]). In contrast, TPSR has substantially lower computation time and the number of equation candidate evaluations, typically in the order of $10^2$ equations, taking approximately $10^2$ seconds (as shown in Fig. 5 and 7). This renders TPSR more suitable for applications where fast yet accurate equation discovery is critical.

### C.3 Caching Details

In the evaluation phase of MCTS, a transformer model is employed to produce complete sequences from a given state. This procedure entails the creation of implicit tree structures that are used to carry out a beam search. The beam search involves determining the *top-k* next tokens for the states visited during the generation process until the entire sequence is generated. These calculations will be needed in future MCTS iterations for two purposes: (1) to extract the *top-k* next tokens during the **expansion** step of each state and (2) to generate the complete equation from a given state during the **evaluation** step. To avoid redundant computations and improve the efficiency of the framework, two caching mechanisms are used, namely *top-k caching* and *sequence caching*.

*Top-k caching* is a mechanism that stores the computed top-k values for given states. For example, in Fig. 4 of the main paper, when evaluating the state $s = [+, \sin]$ in iteration $t$ of MCTS, the *top-k* tokens are calculated for $s$ and its subsequent visited states (e.g., $[+, \sin, x_2]$). These pairs of states and their corresponding *top-k* values can be stored in a *top-k cache*. Consequently, if a state $s$ is visited again in a future iteration (e.g., visiting $s = [+, \sin, x_2]$ in iteration $t + 1$ of MCTS), the cached *top-k* values are utilized instead of calling pretrained SR model again and retrieving the *top-k* tokens from the forward pass of model.

Another mechanism employed to reduce redundant computations is *sequence caching*, which caches complete equations generated in a greedy manner. When the beam size in MCTS is one, the sequence is generated greedily for the given state in the evaluation step. This means that if any partial sequence of this equation is given, the same equation will be generated by the decoder. As a result, the generated equation in iteration $t$ can be used directly in future iterations if the state matches the stored equation partially. For instance, in Fig. 4 of the main paper, consider the equation $\tilde{f} : [+, \sin, x_2, \cdot, x_1]$ is generated for $s = [+, \sin]$ with $b = 1$ in iteration $t$. Now, if in a later iteration (e.g., iteration $t + 1$),

the state to evaluate is $s = [+, \sin, x_2]$, the iterative sequence generation process can be bypassed by directly using the sequence cache to predict the complete equation. It is essential to note that both of these caching strategies serve the same purpose of enhancing the framework's efficiency without compromising its accuracy performance.

## D    Further Results and Visualization

### D.1    Controlling the Fitting-Complexity Trade-off

Fig. 10 illustrates the relationship between fitting accuracy and complexity of predicted equations for various values of $\lambda \in \{0, 1, 0.5, 1\}$, on the *Feynman* dataset. This figure highlights the impact of the controllable complexity parameter $\lambda$ on balancing the trade-off between fitting performance and equation complexity. As it can be observed, when the value of $\lambda$ is set to 0, the TPSR framework generates exceedingly complex equations, resulting in a complexity score greater than 80. These equations are primarily focused on optimizing fitting performance. However, as $\lambda$ is slightly increased to 0.1, there is a minimal effect on the fitting performance, while the complexity of the generated equations drops significantly to a score of less than 60. As $\lambda$ continues to increase, the TPSR framework produces equations with reduced complexity, accompanied by a slight decline in fitting performance. Fig. 10 demonstrates that even when $\lambda$ is set to a large value, such as 1, the fitting accuracy performance of the equations generated by TPSR remains notably superior to the baseline E2E+Sampling method (**0.916** versus **0.848**).

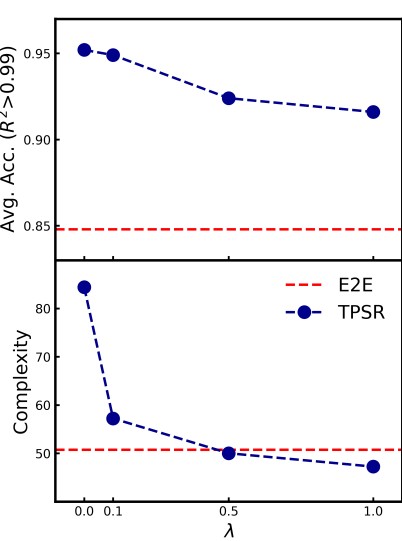

Figure 10:   Effect of controllable complexity parameter ($\lambda$) on average test accuracy and equation complexity for the *Feynman* dataset. E2E uses sampling decoding.

Additionally, the complexity of the generated equations marginally improves (**47.24** compared to **50.73**). This can be observed by examining the gap between the red and blue dashed lines in both the top and bottom sub-figures of Fig. 10. These findings emphasize the advantages of the TPSR framework over the baseline methods in terms of fitting performance. At the same time, TPSR is capable of generating equations with either comparable or lower complexity than those discovered by the baseline methods.

Given the significance of $\lambda$ in governing this trade-off, and to assist users in hyperparameter selection, we recommend setting $\lambda = 0.1$ as a default. Based on our results, particularly Table. 1 and Fig. 10, we find that this setting tends to achieve a harmonious balance between accuracy and complexity, mitigating overfitting. It is important to note that this recommendation aims to offer a starting point for users. The appropriate choice of this hyperparameter may depend on the specific use case, where the balance between finding an accurate function and sacrificing complexity, versus emphasizing interpretability and equation simplicity over relative accuracy, becomes relevant.

### D.2    Pareto Comparisons: Accuracy-Complexity and Accuracy-Time Trade-off

Fig. 11 presents pareto comparisons of various algorithms on two fronts: fitting-complexity (top row) and fitting-time (bottom row) trade-off. These comparisons are conducted on the (a) *Black-box* and *Feynman* datasets. Results show that TPSR demonstrates superior performance, consistently achieving the optimal Pareto-front in all comparisons over both data groups. With respect to the balance between complexity and accuracy (as also noted in Fig. 1), TPSR outperforms the E2E transformer backbone and shows comparable performance to state-of-the-art GP algorithms. TPSR also provides a significant improvement in fitting-time balance, being 100 times faster than leading GP algorithms, a benefit accomplished by utilizing pre-trained priors. Notably, this substantial improvement in inference time does not come at the cost of accuracy, as TPSR also exceeds the E2E baseline as well as most leading GP methods in this aspect.

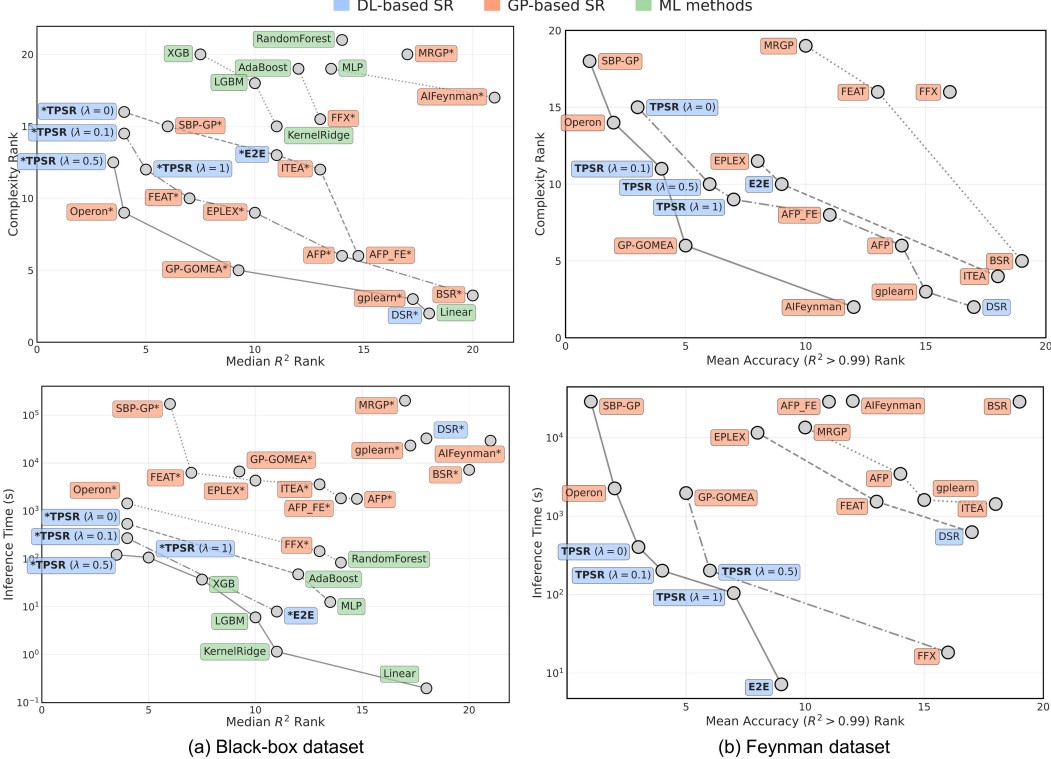

Figure 11: Pareto comparison of all methods in terms of fitting-complexity (top) and fitting-time (bottom) trade-off across **(a) SRBench *Black-box*** and **(b) *Feynman*** datasets. TPSR successfully reaches the first Pareto-front in all comparisons. In terms of fitting-complexity balance, it outperforms E2E baseline, obtaining comparable results to SOTA genetic algorithms. In terms of fitting-time balance, it performs 100x faster than genetic algorithms by leveraging the pre-trained priors and surpasses the accuracy of E2E baseline.

### D.3 Qualitative Study

Fig. 12 offers a detailed qualitative analysis comparing the performance of TPSR, the E2E baseline (symbolic model) as well as XGBoost and MLP (black-box models) with respect to the ground-truth equation $x^2\sin(x)$. The training dataset, depicted by the shaded red region, consists of 200 data points randomly sampled within the range of $(-2, 2)$. The evaluation is performed on an out-of-domain region spanning from $(-5, 5)$.

While all four models demonstrate a strong ability to fit the training data, the proposed TPSR method surpasses the E2E baseline in fitting the true underlying function, as evidenced by its performance in the out-of-domain region. This superior performance can be attributed to TPSR's capacity to generate less complex equations that still effectively fit the data, a feature highlighted in the accompanying complexity barplot. Moreover, the results showcase the general superiority of symbolic regression methods over the black-box XGBoost and MLP machine learning methods when fitting the underlying function within the unseen evaluation range. This observation emphasizes the potential benefits of adopting symbolic regression techniques, such as TPSR, in providing more accurate representations of the data's underlying symbolic patterns and behaviors.

### D.4 Evaluating the Model-Agnostic Capability

In order to underscore the model-agnostic capabilities of TPSR, we also conducted evaluation experiments to include the integration of TPSR with the "Neural Symbolic Regression that Scales" (NeSymReS) model by Biggio *et al.* [16], a pioneering work for large-scale pre-training in SR.

**Limitations and Adjustments.** NeSymReS, while influential, presents some inherent limitations: (1) *Dimensionality Constraint:* It can only handle datasets having a maximum of three dimensions ($D \leq 3$). This limits its application in wider experimental scenarios. (2) *Skeleton Prediction:*

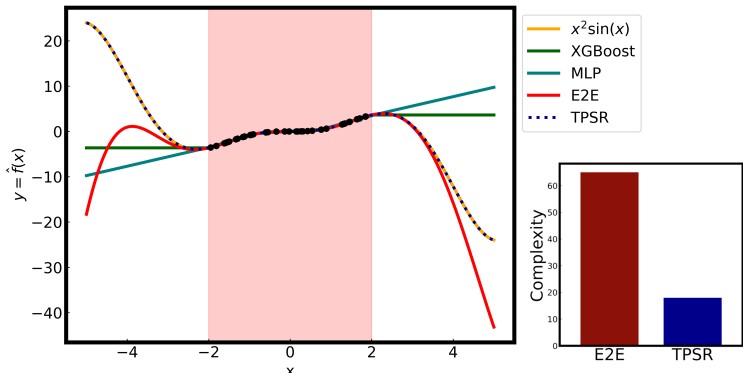

Figure 12: Qualitative comparison of TPSR with E2E as well as black-box XGBoost and MLP models on the ground-truth function $x^2 \sin(x)$. The training dataset contains 200 points in range of $(-2, 2)$ (shaded region), and the performance is evaluated over $(-5, 5)$.

NeSymReS is also trained to only predict equation skeletons. As such, the system requires a more complex constant optimization process, further complicating its integration.

**Experiment Setup.** Due to the constraints highlighted above, to evaluate the combination of TPSR with NeSymReS, we use a dataset composed of $52$ Feynman equations, as in [16], ensuring the dimensionality constraint ($D \leq 3$) is respected.

**Results.** As illustrated in Table 3, integrating TPSR with NeSymReS resulted in marked improvement. Specifically, results show that TPSR has significantly improved the fitting accuracy of NeSymReS without changing the average complexity of the equations when $\lambda = 0.1$ and with a slight increase when $\lambda = 0$.

Table 3: Fitting accuracy and complexity performance of NeSymReS [16] with and without the proposed TPSR planning on $52$ *Feynman* datasets with $D \leq 3$.

| Model | Avg. ($R^2 > 0.99$) $\uparrow$ | Avg. Complexity $\downarrow$ |
|---|---|---|
| NeSymReS | 0.635 | **9.98** |
| NeSymReS+TPSR ($\lambda$=0.1) | 0.808 | **9.98** |
| NeSymReS+TPSR ($\lambda$=0) | **0.827** | 13.30 |

### D.5  Additional SRBench Results

**Strogatz Datasets.**  Fig. 13 presents a performance comparison of TPSR and SRBench algorithms on the *Strogatz* dataset (similar to the results shown for *Feynman* and *Black-box* datasets in Fig. 5). The *Strogatz* dataset comprises 14 equations from a two-state system following a first-order ordinary differential equation (ODE). As it can be observed, E2E performance is less well on this dataset compared to other genetic algorithms due to the unique time-ordered distribution of observations, which differs substantially from the E2E's pre-training data. Notably, despite not being exposed to time-ordered data during pre-training, TPSR with the E2E pre-training backbone significantly enhances its performance on the *Strogatz* dataset. TPSR ranks among the top three baselines for fitting accuracy performance while maintaining comparable or even slightly better equation complexity and inference time.

**Black-box Datasets.**  SRBench [42] studied black-box problems, originally extracted from OpenML [10] and integrated into PMLB [43], include several datasets derived from Friedman's [48] synthetic benchmarks. These Friedman datasets, generated through non-linear functions, display varying degrees of noise, variable interactions, and non-linearity. As observed in earlier studies [42], the results from the Friedman datasets tend to highlight the performance differences among top-ranked methods more noticeably than other benchmarks, where top-performing methods often deliver

---

[10]https://www.openml.org/

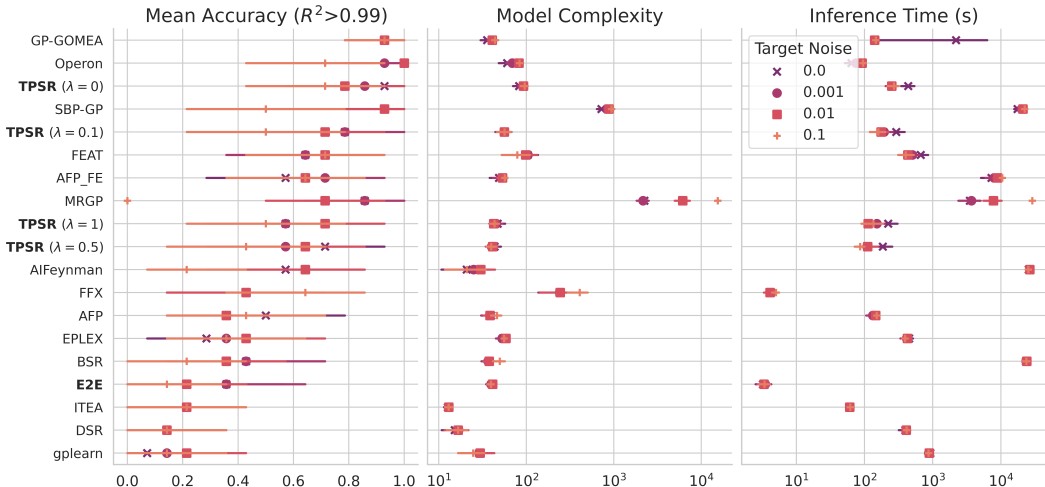

Figure 13: Performance comparison of TPSR and SRBench algorithms in terms of Accuracy-Complexity-Time on *Strogatz* dataset. Models are sorted based on mean accuracy defined as the ratio of solutions with $R^2 > 0.99$ on test set under various noise levels. The error bars represent the $95\%$ confidence interval.

similar results. Fig. 14 shows that performance of several baselines such as KernelRidge, MLP, DSR, BSR, gplearn, and AFP, degrades on Friedman datasets. However, our TPSR variants maintains its superior performance across these challenging Friedman synthetic datasets and the remaining PMLB black-box datasets, asserting its state-of-the-art (**top-**1) status. Following [42], this performance distinction is illustrated in Fig. 14 with more details, separating the results of Friedman datasets from the rest of PMLB black-box datasets.

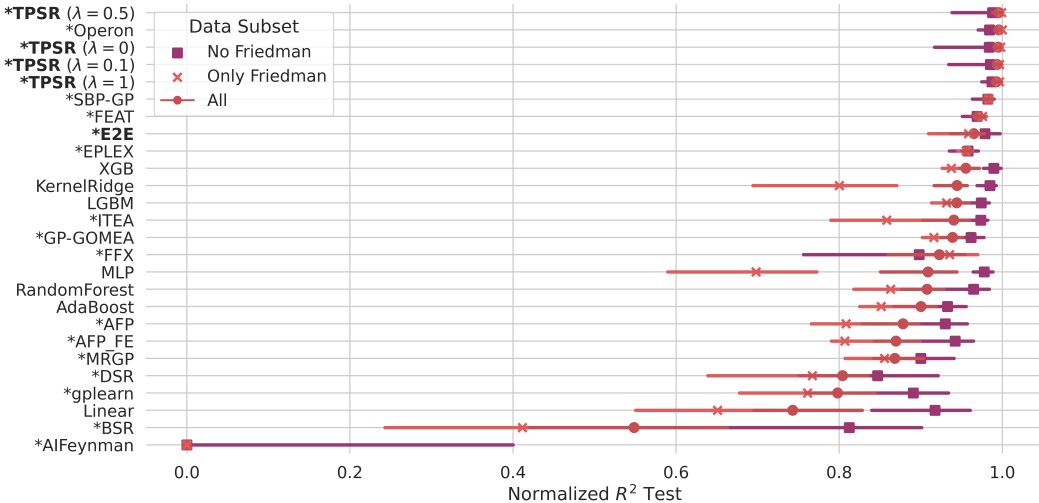

Figure 14: Detailed performance comparison of TPSR and SRBench algorithms in terms of Accuracy (Fitting Performance) on *Black-box* dataset groups: Friedman [48] synthetic datasets, non-Friedman datasets, and all the black-box datasets. The error bars represent $95\%$ confidence interval and " $*$ " refers to SR methods vs. other ML methods.

Fig. 15 shows an in-depth comparison of TPSR performance, varying $\lambda \in \{0, 0.1, 0.5, 1\}$, against top competitors (Operon, SBP-GP, FEAT, EPLEX, and E2E) on *Black-box* datasets of different input dimensions. Given E2E's pre-training on $d_{max} \leq 10$, we focused on datasets with input dimensions $d \leq 10$. In Fig. 15(a), we note that dataset distribution and model performance both depend on the input dimensionality. TPSR consistently outperforms competitors across most dimensions.

Interestingly, lower dimensions (e.g., $d = 3$) favor TPSR with higher $\lambda = 0.5$ or $1$, resulting in better performance, while larger dimensions (i.e., $d = 8, 9$) benefit from smaller $\lambda = 0, 0.1$. This pattern aligns with the expectation that greater $\lambda$ values yield less complex expressions, more prevalent in lower dimensions, and vice versa. Fig. 15(b) presents the average inference time for each model across different input dimensions. E2E is the fastest, while SBP-GP and DSR are the slowest. Notably, as input dimension increases, the inference time of Operon and EPLEX significantly escalates, hitting the scale of $10^4$ and $10^5$ seconds respectively, while TPSR's time remains relatively constant, peaking at $10^3$ seconds or roughly 30 minutes for $d = 9$, compared to Operon's 3 hours and SBP-GP's 30 hours. This shows how efficient TPSR is compared to genetic algorithms in finding higher-quality expressions. Finally, Fig. 15(c) shows the average complexity of expressions generated by each model for different input dimensions. DSR's expressions are the least complex, while SBP-GP's are the most. TPSR with $\lambda = 0$ is slightly more complex than its counterparts. Interestingly, TPSR with $\lambda = 0.5, 1$ produces less complex expressions than GP-based models like Operon, FEAT, and EPLEX at lower dimensions. However, as dimensions increase, these models generate less complex expressions than TPSR.

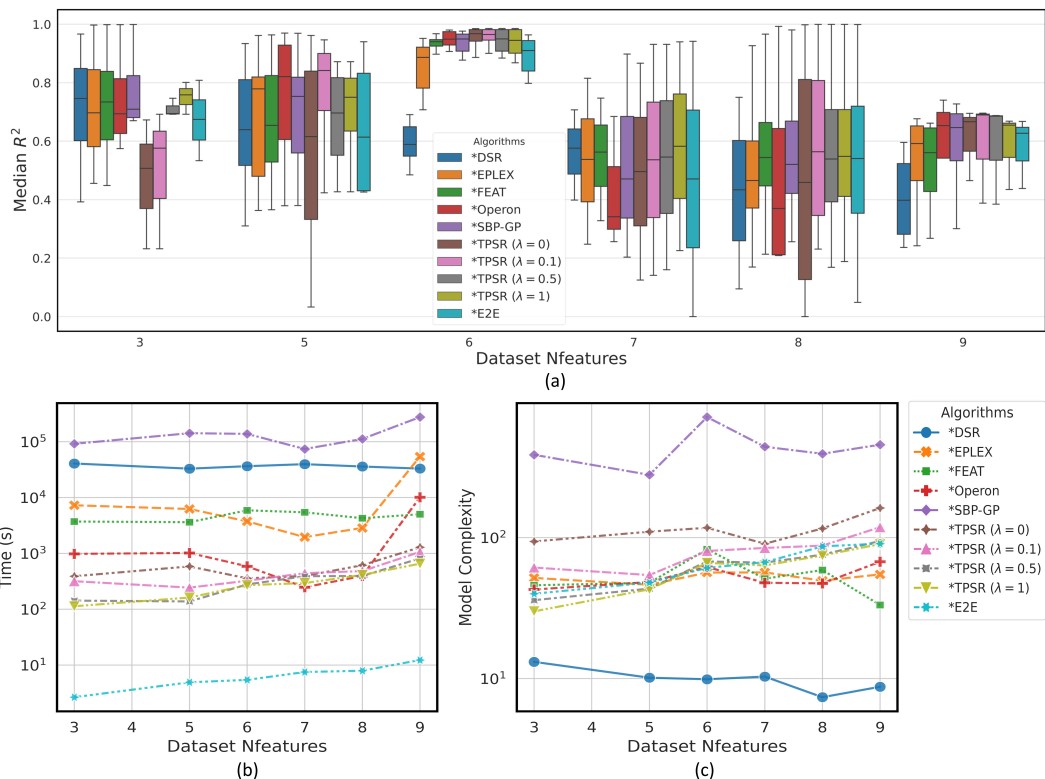

Figure 15: Detailed performance comparsion of TPSR and competing baselines in terms of Accuracy-Complexity-Time metrics for *Black-box* datasets of varying input dimensions.

### D.6    Additional In-Domain Results

Fig. 16 presents a comprehensive performance comparison between our proposed TPSR method with varying controllable parameter $\lambda \in \{0, 0.1, 0.5, 1\}$ and the E2E baseline employing sampling for the *In-domain Synthetic Dataset*. As observed, when the complexity of the synthetic formula increases (as shown in the top row), such as increasing the number of binary/unary operators or the input dimension, the performance across all models tends to degrade. However, we can see that TPSR with $\lambda = 0, 0.1$ "always" have lower performance drops and TPSR with $\lambda = 0.5, 1$ "mostly" have lower performance drop than the E2E. This highlights that not only does the incorporation of performance feedback in TPSR's MCTS-guided decoding help the transformer generation scale better with these difficulty levels, but the controllable complexity parameter $\lambda$ also plays a pivotal role in performance scaling for more challenging input functions.

Fig. 16(d) illustrates that the performance of all models increases as the number of input data points $N$ grows, as one would expect. However, TPSR with $\lambda = 0, 0.1$ exhibits considerably better low-resource performance for $N < 100$ compared to the E2E model. It is important to note that the maximum $N_{max} = 200$ since the E2E model is pretrained with $N \leq 200$, and the transformer architecture employed in the encoding stage demands significant computational and GPU resources for training the model with $N > 200$.

Fig. 16(e) also reveals that the performance of all models improve as the number of input data centroids increases, meaning that as the input data is sampled with greater diversity across different distribution clusters. We can clearly observe that our proposed TPSR with $\lambda = 0, 0.1$ consistently outperforms the E2E model, both with smaller and larger numbers of centroids.

Fig. 16(f) further investigates the impact of introducing multiplicative noise with variance $\gamma$ to the target: $y \rightarrow y(1 + \sigma), \sigma \sim \mathcal{N}(0, \gamma)$. As evident from the figure, the performance of all models deteriorates as the noise variance increases. This phenomenon highlights the sensitivity of the pre-trained models to the input noise of the target variable. However, it is noteworthy that TPSR with $\lambda > 0$ demonstrates slightly better performance compared to the E2E model, particularly when encountering larger noise variances.

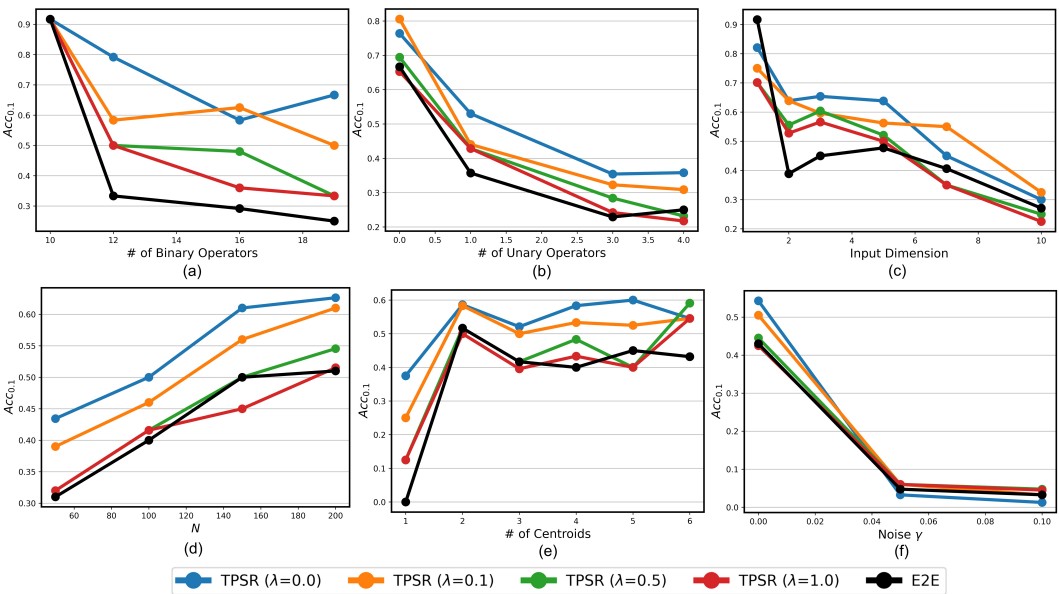

Figure 16: Performance comparison of TPSR for varying $\lambda \in \{0, 0.1, 0.5, 1\}$ and E2E with sampling decoding across different levels of formula and input difficulties: **(a) number of binary operators, (b) number of unary operators, (c) input dimension, (d) number of input points $N$ (e) number of input centroids, and (f) input noise variance $\gamma$.**

## D.7 Additional Ablation Studies

The selection of $\beta$, in Eq. (1) can also affect the exploration-exploitation trade-off, influencing the overall performance of TPSR. Fig. 17 demonstrates the impact of varying $\beta$ on TPSR's performance over 119 *Feynman* datasets, emphasizing the balance between exploration and exploitation. Based on the results, we observe that for small values of $\beta$, specifically $\beta = 0$, the performance is sub-optimal. This diminished performance can be attributed to constrained exploration. Without sufficient exploration, the model might miss potential solutions or equation sequences that might be more effective. At the other end of the spectrum, with large values like $\beta = 100$, there is also a decline in performance. This degradation can be linked to an over-emphasis on exploration at the cost of exploitation. By exploring too much without adequately leveraging the learned knowledge, the model can get overwhelmed with possibilities, some of which might not be beneficial. Experiment results highlight that optimal performance is achieved for $\beta$ values ranging between 0.1 and 10. As seen in Fig. 17(b), with an increase in $\beta$, the number of equation sequence candidates grows, indicating an increase in exploration. However, beyond $\beta > 0.1$, the increase in sequence candidates

is marginal. This plateau suggests the possible activation of caching mechanisms due to repetitive sequence generation. Fig. 17(a) also shows average accuracy performance against different $\beta$ values, illustrating the aforementioned trends and offering a visual guide for selecting $\beta$.

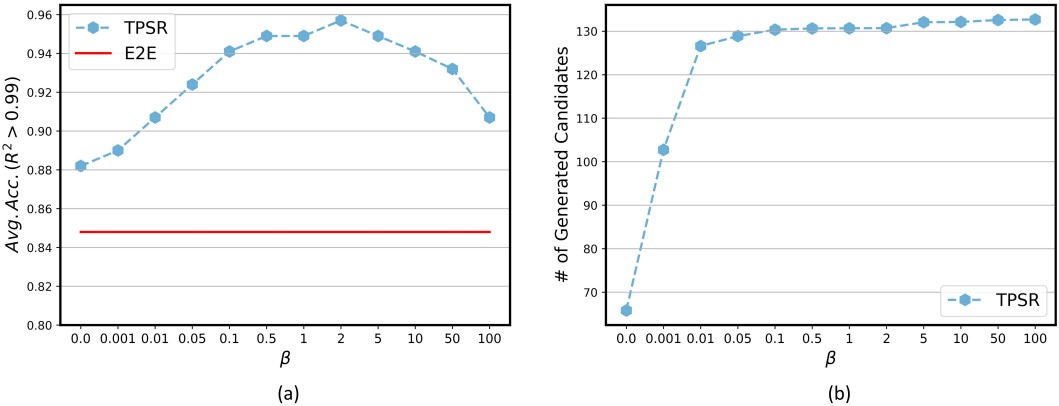

(a) (b)

Figure 17: Ablation study of $\beta$ parameter in TPSR on 119 *Feynman* datasets: Balancing Exploration and Exploitation.

### D.8 Examples of Generated Symbolic Expressions

Table 4 presents example comparisons of symbolic expressions generated by E2E using sampling and our proposed TPSR model with $\lambda = 1$ for 200 observation points of given true functions. To improve readability and simplify notation, all constants in the generated expressions are denoted with the token "C". The table highlights how TPSR-generated symbolic expressions are more closely aligned with the true functions than those generated by E2E. The aligned components are bolded in the table entries. Additionally, the fitting performance $R^2$ of TPSR-generated equations is notably superior to that of E2E-generated expressions. This comparison demonstrates how TPSR's integration of fitting and complexity feedback during transformer decoding can yield quantitatively and qualitatively improved expressions using the same model weights. Improved learning of governing expressions can enhance the interpretability of black-box prediction models, contributing to their extrapolation and generalizability.

Table 4: Example comparisons of symbolic expressions generated by E2E and TPSR, along with their respective fitting performance.

| | Expression | $R^2$ |
|---|---|---|
| True Function | $2X_0(1 - \cos\left(\boldsymbol{X_1 X_2}\right))$ | – |
| E2E Generation | $CX_0\left(C + C\cos\left(CX_2 + CX_1\right)\right) + C$ | 0.453 |
| TPSR Generation | $CX_0\left(C + C\cos\left(CX_1 + CX_2 + \boldsymbol{CX_1 X_2}\right)\right) + C$ | **1.0** |
| True Function | $\boldsymbol{\sin^2\left(\frac{X_0 X_1}{\left(\frac{X_2}{2\pi}\right)}\right)}$ | – |
| E2E Generation | $C\sin\left(CX_0 + CX_1 + CX_2\right) + C$ | 0.178 |
| TPSR Generation | $C\boldsymbol{\sin^2}\left(\frac{CX_1 X_0}{CX_2 + CX_1}\right) + C$ | **0.671** |
| True Function | $X_0\left(\cos\left(\boldsymbol{X_1 X_2}\right) + \boldsymbol{X_3}\cos^2\left(\boldsymbol{X_1 X_2}\right)\right)$ | – |
| E2E Generation | $CX_0\left(CX_3 + CX_2 + CX_1 + C\cos\left(CX_2 + CX_1\right)\right)^2 + C$ | 0.878 |
| TPSR Generation | $CX_0\left(\cos\left(\boldsymbol{CX_2 X_1}\right) + X_3^2\cos^2\left(\boldsymbol{CX_2 X_1}\right)\right) + C$ | **0.996** |
| True Function | $X_0\frac{\sin^2\left(\frac{X_1 X_2}{2}\right)}{\sin^2\left(\frac{X_2}{2}\right)}$ | – |
| E2E Generation | $CX_0 + CX_0\left(C\sin\left(CX_1 + CX_2\right) + CX_1^2\right)^2 + C$ | 0.655 |
| TPSR Generation | $CX_0^2\left(\frac{\sin\left(CX_2 X_1\right)}{\sin\left(CX_2\right)}\right)^2 + C$ | **0.991** |
| True Function | $\sqrt{\left(X_0^2 + X_1^2 - 2X_0 X_1\cos\left(\boldsymbol{X_2 - X_3}\right)\right)}$ | – |
| E2E Generation | $\sqrt{\left(CX_0 + CX_1\right)^2\cos\left(CX_2 + CX_3^2\right)} + C$ | 0.939 |
| TPSR Generation | $\sqrt{\left(CX_1 X_0\cos\left(\boldsymbol{CX_2 - CX3}\right) - CX_1^2 X_0^2\right)} + C$ | **0.986** |

# E   Discussion and Future Work

**Limitations.**    While our methodology exhibits substantial potential, it is not without limitations. One limitation of our approach is the increased inference time of the TPSR in comparison to simpler decoding methods like beam search and sampling. This extended inference time is primarily due to the process of searching and incorporating performance feedback during the generation phase in TPSR's decoding process. Nevertheless, by exploiting the large-scale pre-trained priors, TPSR's inference time still remains considerably lower than the majority of leading genetic algorithms. Another factor influencing TPSR's performance is the dependency on the learned priors of the pre-trained transformer model. TPSR is also subject to the inherent structural limitations of the pre-trained model, such as constraints on input dimensionality, expression length, and vocabulary definition. For example, the E2E model is pre-trained with a maximum input dimension ($d_{max}$) of 10, which in turn limits the TPSR with the E2E backbone to $d \leq 10$. However, it's important to note that TPSR is a model-agnostic framework, implying potential integration with more advanced pre-trained SR models in the future.

**Future Directions.**    An intriguing dimension in the symbolic regression revolves around out-of-distribution data. Pre-trained Transformer SR methods, distinct from their search-focused counterparts, train on vast synthetic datasets stemming from certain distributions. Essentially, this distribution is shaped by specific equation generators and sampling techniques. Hence, any data or equation not stemming from these generators could be viewed as out-of-distribution. Our experimentation evaluated TPSR and the pre-trained E2E model [18] across both in-domain and out-of-distribution datasets, as in the SRBench. A crucial observation was that TPSR, with lookahead planning, considerably elevates the pre-trained model's performance on out-of-distribution datasets, a trend most pronounced in SRBench comparisons (as illustrated in Table 1). While pre-trained models offer the strength of utilizing prior knowledge from large-scale datasets, they can be limited when faced with data far from their training distribution or unique equation forms they have not encountered during training. TPSR offers a partial solution through its decoding-stage search and lookahead planning, but it is still limited to the inherent constraints of the pre-trained SR model's priors. Addressing this challenge is an intriguing avenue for future research. Possible strategies might involve fine-tuning pre-trained model weights using non-differentiable rewards for the new out-of-distribution datasets.

# F   Broader Impacts

**Potential positive impacts.**    The proposed TPSR approach for symbolic regression using transformer-based models has significant implications for both the research and practical communities. By integrating Monte Carlo Tree Search (MCTS) into the transformer decoding process, TPSR enables the generation of equation sequences that balance fitting accuracy and complexity, addressing key challenges in symbolic regression. This has wide-ranging applications in science and engineering domains, where accurate and interpretable mathematical models are essential for understanding and predicting complex phenomena. The improved performance of TPSR over state-of-the-art methods enhances the usability and reliability of symbolic regression models, enabling researchers and practitioners to extract valuable insights from their data and make informed decisions.

Moreover, TPSR offers practical benefits by leveraging the efficiency of transformer-based models and the pre-training priors. The ability to optimize equation generation using TPSR enhances the efficiency and scalability of symbolic regression, making it more accessible in resource-constrained settings. This opens up opportunities for the adoption of symbolic regression in various domains, including scientific research, engineering design, and optimization problems. The impact of TPSR extends beyond symbolic regression, as the integration of MCTS and non-differentiable feedback into transformer-based models can inspire novel approaches in other fields where the combination of symbolic mathematical or formal verification and reasoning with machine learning is valuable. Overall, TPSR has the potential to advance the state-of-the-art in symbolic regression and contribute to scientific and technological advancements.

**Ethical considerations.**    Symbolic regression makes it easier for anyone to understand underlying symbolic and mathematical patterns behind the data and learn interpretable mathematical models for observations. This approach brings the potential for machine learning models to achieve a balance of high predictive performance and transparency, which is critically valuable in sectors such as healthcare, where the interpretability of models can directly influence life-saving decisions. However,

as with any powerful tool, the ethical issues of its use must be considered carefully. For example, while symbolic regression can yield life-saving insights in the hands of healthcare professionals, it can also be exploited for malicious purposes. It could be used to decipher patterns and relationships within data where privacy should be maintained, leading to potential breaches of confidentiality. This becomes particularly concerning as symbolic regression techniques mature, enabling more effective comprehension of symbolic mathematical and causal relationships behind data values. To mitigate this risk, we need the development of separate modules tasked with screening input data and denying requests where pattern extraction could lead to harmful outcomes.