# OpenReview forum: "Transformer-based Planning for Symbolic Regression"
_NeurIPS.cc/2023/Conference — NeurIPS 2023 poster_

### Official Review · Reviewer_ijKU · 2023-07-02

**Soundness:** 3 good
**Presentation:** 3 good
**Contribution:** 2 fair
**Rating:** 6
**Confidence:** 4

**Summary:**

The paper introduces TPSR, a Transformer-based Planning strategy for Symbolic Regression. TPSR incorporates Monte Carlo Tree Search into the transformer decoding process, enabling the integration of non-differentiable feedback such as accuracy and complexity. Experimental results show that TPSR outperforms existing methods in terms of fitting-complexity trade-off, extrapolation abilities, and robustness to noise.

**Strengths:**

- The paper is well written and easy to understand.
- The idea of enhancing large scale pre-trained Transformers with improved search capablities is very promising in the context of symbolic regression
- The model shows good performance both compared to the E2E baseline and the GP methods.

**Weaknesses:**

- My main concern is about the novelty of the approach. A very similar idea has been recently investigated in [1] where the authors also combine MCTS with pre-trained Transformers. I would be grateful if the authors could clarify any eventual differences between the two approaches.
- The impact of $\lambda$ seems quite significant in your experiements. However, it is not clear to me how one should select it in practice.


[1] Kamienny, Pierre-Alexandre, Guillaume Lample, and Marco Virgolin. "Deep Generative Symbolic Regression with Monte-Carlo-Tree-Search." (2023).

**Questions:**

Please refer to the weakness part above.

---

> ### Author Rebuttal · Authors · 2023-08-09
>
> Thank you for the valuable feedback on our work. We appreciate your positive comments on the clarity and potential of our work in symbolic regression.
>
> ---
> > * My main concern is about the novelty of the approach. A very similar idea has been recently investigated in [1] where the authors also combine MCTS with pre-trained Transformers. I would be grateful if the authors could clarify any eventual differences between the two approaches.
> >
>
> We understand your concern about the novelty of our approach and its potential similarity to the work [1]. Allow us to clarify the distinct differences between our Transformer-based Planning for Symbolic Regression (TPSR) and the DGSR-MCTS approach:
>
> * **General Approach:** The general mechanism used for generating the equations is different in DGSR-MCTS and TPSR. DGSR-MCTS exploits a pretrained mutation policy M to generate the expression by following a series of mutations from an empty expression (root). This is while TPSR follows the seq2seq approach of E2E to generate the expression token-by-token. Consequently, TPSR uses the pretrained E2E as its backbone but DGSR-MCTS pretrains the mutation policy from scratch.
>
> * **Definition of MCTS and Search Strategy:** One fundamental distinction lies in the definition and application of Monte-Carlo Tree Search (MCTS). In DGSR-MCTS, the search tree consists of full mathematical equations, with each node representing a distinct equation and edges corresponding to mutations between equations. In contrast, our TPSR employs MCTS as a decoding strategy in the context of the transformer model. Each node in the search tree of TPSR represents the current state of generated tokens, potentially forming non-complete sequences, with edges corresponding to mathematical operators or variables. As a result, the search tree of DGSR-MCTS with "n" nodes includes "n" different equations, while the TPSR search tree includes intermediate decoding sequences, and completed equations only exist at the terminal nodes. This distinction inherently leads to major differences in selection, expansion, and back-propagation mechanisms within the MCTS algorithm.
>
> * **Parameter Update and Learning:** DGSR-MCTS utilizes MCTS to update and learn the distribution of mutations for a group of out-of-distribution datasets. The approach involves fine-tuning an actor-critic-like model to adjust the pre-trained model on a group of symbolic regression instances. On the other hand, TPSR uses the pre-trained transformer's learned distribution to guide the expansion during the search process, without updating any specific parameters for in-domain or out-of-domain equations (without fine-tuning). Consequently, the same settings and pre-trained model are applied to both in-domain and out-of-domain equations in TPSR.
>
> * **Computation Time:** Another notable difference is the computational requirements of the two approaches. DGSR-MCTS involves pre-training a mutation policy, a critic network, and performing fine-tuning stages for these networks, leading to significantly higher computation time (a limit of 24hrs and 500K evaluations as stated in their original paper). In contrast, TPSR has substantially lower computation time and the number of evaluations, typically in the order of $10^2$ equations, taking approximately $10^2$ seconds (as shown in Fig. 6 and 7 of the main paper). This renders TPSR more suitable for applications where fast yet accurate equation discovery is critical.
>
> [1] Pierre-Alexandre Kamienny, Guillaume Lample, Sylvain Lamprier, and Marco Virgolin. "Deep Generative Symbolic Regression with Monte-Carlo-Tree-Search." (2023).
>
> ---
> > * The impact of $\lambda$ seems quite significant in your experiements. However, it is not clear to me how one should select it in practice.
> >
>
> Indeed, the impact of $\lambda$ as the complexity regularizer is significant in most cases. We evaluated and discussed different values of $\lambda$ as it can affect the trade-off between accuracy and complexity, as shown in Table 1 and Fig. 10 (Appendix D.1). The appropriate choice of this hyperparameter may depend on the specific use case, where the balance between finding an accurate function and sacrificing complexity, versus emphasizing interpretability and equation simplicity over relative accuracy, becomes relevant. However, we agree that proposing default settings for hyperparameters would be beneficial for general use. Based on our results, particularly Fig. 10 in Appendix D.1, we conclude that setting $\lambda = 0.1$ can achieve high accuracy while reducing complexity and avoiding overfitting (please also check Fig. 6). We will make sure to include this discussion in the updated manuscript.

---

> > ### Comment · Reviewer_ijKU · 2023-08-16
> >
> > Thank you for your response to my review.
> >
> > I have raised the rating after reading the rebuttal. I would suggest the authors update the manuscript to better clarify the above points.

---

> > > ### Author Response · Authors · 2023-08-16
> > > **Thank you**
> > >
> > > Thank you for reviewing our rebuttal. We are glad that our response has resolved your concerns and appreciate the raised score. We will make sure to update the manuscript accordingly and include the above points in the updated version.

---

> ### Author Response · Authors · 2023-08-14
> **Looking forward to discussion**
>
> Dear Reviewer ijKU,
>
> Thank you for your feedback during the review process! If there are any concerns or questions, please do not hesitate to let us know - before the author discussion period ends. We will be happy to answer them during the discussion.
>
>
> Thank you,
>
> Paper13018 Authors

---

### Official Review · Reviewer_QJmi · 2023-07-04

**Soundness:** 3 good
**Presentation:** 3 good
**Contribution:** 3 good
**Rating:** 7
**Confidence:** 3

**Summary:**

Authors propose a transformer-based planning (using MCTS) strategy to solve symbol regression task. Different from traditional decoding method, the new method is able to integrate non-differentiable feedback into the transformer-based process of equation generation. Experiments demonstrate the significent performance.

**Strengths:**

Distilling symbolic equation from noisy data is intractable. Recent progress is achieved by training neural networks to generate candidate symbolic expressions, which is really promising.

This work combines the Monte Carlo Tree Search and pretrained transformer-based symbol regression model for equation generation. Compared with Genetic programming method, the new approach not only leverages pre-trained priors, but also considers feedbacks during the generation process.

**Weaknesses:**

There is not much initiality in the new method. It demonstrates a new application for a combination of two existing methods.

**Questions:**

Are there experiments to show changes of performances, if we change the selection set of mathematical operators and symbols?

Can out-of distribution data be identified, and used for the promotion of the symbolic regression process?

It might not be diffiuclt for symbol regression method to find laws, such as f=ma. Could it find E = m c^2?

**Limitations:**

Monte Carlo Tree Search is statistical, and Pre-trained transformer is trained through data, the integration of the two methods is still within the traditional paradigm of machine learning, so, may not work well for out-of distribution data.

---

> ### Author Rebuttal · Authors · 2023-08-09
>
> Thank you for the insightful comments and questions on our paper. We appreciate your positive remarks regarding the importance of this study.
>
> ---
> > * Are there experiments to show changes of performances, if we change the selection set of mathematical operators and symbols?
> >
>
> In symbolic regression, the choice of mathematical operators and symbols significantly impacts the equations obtained. Our work proposes a transformer-based decoding strategy with MCTS, which requires adhering to the vocabulary of operators defined by the pre-trained transformer SR model (As mentioned in Appendix E, limitations, line 768). This constraint ensures compatibility with the pre-trained model but may restrict the set of available mathematical symbols.
>
> Generally, the selection of mathematical symbols in symbolic regression involves a trade-off between expressivity and problem complexity. Larger vocabularies provide greater expressivity, allowing the method to represent more diverse equations. However, this increase in expressivity can also enlarge the search space, making the problem more complex. To strike a balance, most recent symbolic regression works use common mathematical operators that are prevalent in benchmark problems and scientific datasets, such as the Feynman dataset.
>
>
> ---
> > * Can out-of distribution data be identified, and used for the promotion of the symbolic regression process?
> >
>
> We believe that this is a very important question in symbolic regression, especially regarding pretrained SR models. We would like to discuss this from two aspects.
>
> First, we would like to discuss the comparison of our TPSR with pretrained models. Pretrained symbolic regression methods, in contrast with search methods, are trained with a large set of synthetic equations, originating from a distribution $\Omega$. In fact, these equations are generated using an expression generator with specific settings, and the points are sampled in specific ways. Therefore, equations and/or datasets that are not generated from the same generator can be considered out-of-distribution. In our work, we evaluate our TPSR model and the pretrained E2E model on both in-domain equations from the same distribution of data, as well as SRBench equations which are considered out-of-distribution compared to the training samples. As shown in Table 1 of the main paper,  using lookahead planning in TPSR significantly improves the performance of the pretrained E2E model on out-of-distribution SRBench datasets. We have also observed that the performance improvement gap between TPSR and E2E is higher for SRBench datasets compared to this gap in in-domain datasets (also discussed in lines 271-275).
>
> Second, from a broader perspective, since pretrained models have learned parameters conditioned on the input datasets and equations (in comparison to search methods), while they have the advantage of leveraging priors learned from large-scale data, they are limited in handling datasets that are very far from the training distribution and discovering very different equation forms from what was generated during the training. TPSR can help to remedy this issue by searching and lookahead planning in the decoding stage; however, it is still limited to the degree that it can be applied to out-of-distribution datasets. This is because of the dependency of TPSR on the fixed priors of the pretrained SR model. We believe that improving SR models for this purpose is an exciting line of research that should be considered for future works, as also mentioned in our conclusion section (lines 360-362). Some potential ideas include fine-tuning the weights of the SR model using non-differentiable rewards for the new out-of-distribution datasets to improve its performance.
>
> ---
> > * It might not be diffiuclt for symbol regression method to find laws, such as f=ma. Could it find E = m c^2?
> >
>
> Thank you for raising this thought-provoking question. We believe that this question can be examined from various angles, and we have tried to address your concerns below.
>
> Symbolic regression methods operate by fitting equations to datasets of observations (features X and corresponding y). When considering the applicability of these methods, it's important to acknowledge that the original benchmark datasets, including well-known cases like the Feynman dataset, often do not span extreme ranges of values for X. As an example, Equation I.48.20 from the Feynman dataset ($e = \frac{mc^2}{\sqrt{1-\frac{v^2}{c^2}}}$) samples values in the range of $U(1,5)$ for $m$, $U(3,10)$ for $c$, and $U(1,2)$ for $v$. This is while, for instance, the value of $c$, representing the speed of light, could be considered a constant with a significantly higher value of $2.998*10^8$ which is far from the simplified covered range in Feynman dataset. In fact, when operating under simplified assumptions and utilizing large quantities of synthetic observations, it is possible to recover complex equation forms using advanced symbolic regression models. However, real-world scenarios present greater complexity due to factors such as diversity, ranges, and precision of observations.
>
> Besides the range of observations, situations like the equation $E=mc^2$ pose a challenge when assuming $c$ to be a constant. In such cases, identifying $c$ requires additional constraints, such as employing dimensional analysis. Notably, due to the inherent constant nature of $c$, even if a diverse dataset encompassing various value ranges for $m$, $c$, and $E$ is employed, the relationship between $m$ and $e$ would exhibit a linear correlation. Accordingly, we recognize that there are inherent limitations in scientific discovery when using contemporary symbolic regression models. Addressing these limitations offers an exciting avenue for future advancements in the field.
>
> We hope our response addresses your concern. Please do let us know if we have correctly interpreted your concern or if further clarification is needed.

---

> ### Author Response · Authors · 2023-08-14
> **Looking forward to discussion**
>
> Dear Reviewer QJmi,
>
> Thank you for your feedback during the review process! If there are any concerns or questions, please do not hesitate to let us know - before the author discussion period ends. We will be happy to answer them during the discussion.
>
>
> Thank you,
>
> Paper13018 Authors

---

### Official Review · Reviewer_6B7y · 2023-07-05

**Soundness:** 3 good
**Presentation:** 4 excellent
**Contribution:** 3 good
**Rating:** 7
**Confidence:** 4

**Summary:**

This paper proposes to incorporate  Monte Carlo Tree Search (MCTS) on top of pretrained transformer-based SR models to guide equation sequence generation. This is to address the challenges where existing methods purely rely on the pretrained transformer’s output and without accounting for external performance requirement. In MCTS, the authors develop a reward function to encourage the balance between fitting accuracy and regulating complexity for the SR generation. Also, the caching tricks are employed to improve the implementation efficiency. SR benchmark datasets are used to demonstrate the improved performance of the proposed method over the state-of-the-art.

**Strengths:**

Including performance feedback in the pipeline of generation of SR equation generation from pre-trained transformer-based SR models is well-motivated.  To achieve so, this paper proposes including MCTS as the decoder in this pipeline, and imparting    the external requirement, via a reward function in MCTS, to eventually improve the performance of equation generation. The extensive experiments and baseline comparison clearly show the effectiveness of the proposed method, in terms of fitting-complexity trade-off, extrapolation abilities, and robustness to noise.

The presentation of techniques is clear, and the evaluation in my opinion is solid. Overall, this paper makes a good contribution in the SR field.

**Weaknesses:**

I only have two comments:

- In Equ. (1), how to select $\beta(s)$? It would be better to show its effect on the performance in the ablation study as well.
- Currently, the method still relies on a pre-trained transformer SR model. The authors could give some perspective about how (or if it is possible) the MCTS can also be incorporated in the transformer training (or fine-tuning) process.
-A typo in line 151: trnasformer--> transformer

**Questions:**

See my comments in the above section.

**Limitations:**

See my comments in the above section.

---

> ### Author Rebuttal · Authors · 2023-08-09
>
> We appreciate your recognition of our motivation and contribution. The typo that you raised has been corrected and the response to your comments is provided below.
>
> ---
> > * In Equ. (1), how to select $\beta(s)$? It would be better to show its effect on the performance in the ablation study as well.
> >
> In response to your comment, we conducted ablation study with different values of $\beta(s)$ on the Feynman SRbench datasets. Our findings, illustrated in Fig. 1 of the response PDF (please check global response), show: (1) small values of $\beta(s)$ (e.g., $\beta(s)$=0) offer lower performance, probably due to limited exploration. (2) large values of $\beta(s)$ (e.g., $\beta(s)$=100) also affect performance negatively due to excessive exploration compared to exploitation. (3) Optimal results are seen for $\beta(s)$ between 0.1 and 10. Also, Fig. 1(b) shows that as $\beta(s)$ increases, more equation sequence candidates emerge, signaling more exploration. However, after $\beta(s)$>0.1, the candidate count doesn't grow much, possibly due to repetitive sequences frequently activating caching mechanisms. We will make sure to include this ablation study in the updated version of paper.
>
>
> ---
> > * Currently, the method still relies on a pre-trained transformer SR model. The authors could give some perspective about how (or if it is possible) the MCTS can also be incorporated in the transformer training (or fine-tuning) process.
> >
>
> We thank the reviewer for raising this very interesting question! Incorporating MCTS into the training/fine-tuning of the transformer SR models is indeed an intriguing direction for future work (we have also briefly mentioned this in the conclusion section (Lines 361-362)). To train/fine-tune transformer SR models with non-differetiale equation-specific feedback, one way is to employ deep reinforcement learning techniques such as policy gradient so that we can backpropagate to update model weights. However, if we want to use the search ability of MCTS instead of RL and policy gradient methods, following these directions might help: **(1)** Utilizing the transformer SR model as both the policy and value network. This actor-critic process can predict action probabilities, hence guiding MCTS to more promising search paths. Additionally, it can estimate the value of specific states, assisting MCTS during reward backpropagation. **(2)** Leveraging trajectories generated from MCTS, or its rollouts, to enrich the training set for the transformer model. These trajectories, particularly the novel solutions discovered during MCTS explorations, can expose the model to non-obvious equation formulations it might not have encountered in traditional training sets. **(3)** Crafting a co-adaptation feedback loop between MCTS and the transformer model where MCTS and the transformer model parameters can be adjusted iteratively based on each component's performance feedback.
>
> We would like to point out that these potential directions will certainly involve intricate challenges and may demand rigorous experimentation to ascertain their efficacy. We acknowledge the constructive nature of your comment, and it further strengthens our resolve to delve into this direction in future works.

---

> ### Author Response · Authors · 2023-08-14
> **Looking forward to discussion**
>
> Dear Reviewer 6B7y,
>
> Thank you for your feedback during the review process! If there are any concerns or questions, please do not hesitate to let us know - before the author discussion period ends. We will be happy to answer them during the discussion.
>
>
> Thank you,
>
> Paper13018 Authors

---

### Official Review · Reviewer_zZmW · 2023-07-06

**Soundness:** 3 good
**Presentation:** 3 good
**Contribution:** 2 fair
**Rating:** 6
**Confidence:** 4

**Summary:**

This submission proposes a neural network-based approach to symbolic regression (SR), namely generating equations as sequences. It leverages the power of pretrained SR transformer models and the MCTS algorithm to tradeoff the fitting accuracy and equation complexity. Experimental results on the SRBench and the In-domain Synthetic datasets demonstrate that the proposed approach outperforms the backbone E2E transformer model.

**Strengths:**

Soundness:
The techniques employed in the proposed approach are sound. The approach is able to use any non-differentiable target function to guide the training of a neural model for symbolic regression. Experiments demonstrate that it outperforms a state-of-the-art transformer model which is used as the backbone in the proposed approach, indicating that the implementation of the proposed approach is likely to be correct.

Presentation:
The submission is in general well written and organized, easy to follow.

**Weaknesses:**

Presentation:
There is a minor issue on the term single-instance symbolic regression introduced in Related Work. According to the description about the therein algorithms GP, RL, GP+RL and MCTS, the difference between them and the proposed approach mainly lies in not employing pretrained knowledge. Thus, the term single-instance is strange and cannot tell the true difference from the proposed approach.

Contribution:
The proposed approach seems to be a combination of the E2E transformer model [18] and the MCTS framework [26]. Although the transformer model can be replaced with other neural network-based models, the contributions beyond [18] and [26] are not significant. Moreover, the current evaluation cannot confirm that either the proposed approach is a general framework for enhancing any neural network-based model for symbolic regression, or the approach achieves the truly state-of-the-art performance. For the former confirmation, the authors need to compare multiple implementations having different backbone models with the original backbone models. For the latter confirmation, the authors need to compare the proposed approach with more state-of-the-art solutions such as [30] and [31].


**Questions:**

Why the algorithms GP, RL, GP+RL and MCTS used for symbolic regression are called single-instance?

**Limitations:**

As far as I can see, the authors have adequately addressed the limitations through sufficient discussions in the supplemented material.

---

> ### Author Rebuttal · Authors · 2023-08-09
>
> Thank you for your valuable review of our submission. We appreciate your positive feedback and would like to address your concerns.
>
> **Terminology: Single-Instance SR.** We acknowledge that the distinction might not be clearly conveyed by the term "single-instance SR" itself. The intention behind using this term was to highlight that algorithms like GP and RL for SR typically focus on finding the best-fit equation for a "single" dataset at hand, without leveraging pretrained knowledge from a large set of datasets. However, we understand that this term may not fully capture the differentiation from the methods discussed in the next paragraph of Related Work which use pretrained knowledge for SR. To enhance clarity, we will revise this and use "symbolic regression without learned priors" instead of "single-instance symbolic regression". We hope that this clarification addresses your concern. If you have any further suggestions or insights, we would appreciate your input.
>
> **Contribution.** While we integrate both pretrained SR models and MCTS, our contribution goes beyond merely combining two existing techniques. The MCTS planning employed in our work takes inspiration from similar methods used in NLP, such as [34], where MCTS is applied for text generation. However, utilizing MCTS planning for equation generation in SR introduces significant differences from its application in NLP. Incorporating the MCTS planning algorithm into the pretrained SR backbone demands a thorough redesign and the introduction of novel techniques to effectively address the distinctive challenges posed by this integration. We firmly believe that this differentiation constitutes an innovative aspect in itself (refer to Appendix C.2 and C.3, as well as Figures 8 and 9).
>
> We want to again emphasize the following contributions:
>
> * We are the first to combine MCTS as a planning-based decoding module with pretrained SR models for the task of equation generation. TPSR not only achieves significantly better performance than the backbone pretrained model but also holds competitive performance compared to the established baselines.
> * We have designed the interfaces between these two components (pretrained SR model and planning search) effectively and dealt with unique challenges of making the framework computationally more efficient.
> * We have also showcased the versatility of TPSR across different objectives, from fitting accuracy to complexity. Notably, TPSR allows for optimizing equation learning based on varying objectives without necessitating finetuning of the large pretrained SR models.
>
> **Model-Agnostic and SOTA Confirmation.**
> In the current version of the manuscript, we have explicitly indicated that our framework is model-agnostic and holds the potential to enhance sequence generation in a variety of pretrained SR models. This includes both existing SR models and potential future models that might exhibit greater capabilities. However, we originally provided the results of using TPSR only on the E2E backbone [18], as E2E is the SOTA pretrained SR model with open source code, and publicly accessible model's weights and logits. In response to your valuable suggestions, we explored integrating TPSR with other pretrained SR backbones to illustrate its model-agnostic enhancement capabilities. Consequently, we integrated our TPSR planning strategy with "Neural Symbolic Regression that Scales" (NeSymReS) by Biggio et al. [16], a pioneering work that proposes large-scale pretraining for SR. However, NeSymReS does have some limitations, including its acceptance of datasets with a maximum of three dimensions. Also, this approach predicts equation skeletons and requires a more complex constant optimization process. Despite these challenges, we evaluated both NeSymReS and its TPSR-enhanced version using a dataset comprising 52 Feynman equations with a dimensionality of $d \leq 3$. Results are provided in Table 1 of the response PDF (please check global response), showing that TPSR has significantly improved the fitting accuracy without changing the average complexity of the equations when $\lambda$=0.1 and with a slight increase when $\lambda=0$. We will include these results in the appendix and refer the readers to that section when discussing the model-agnostic feature of our framework. We would like to remark that due to the limitation of NeSymReS to very low-dimension problems, it cannot be evaluated on the SRBench datasets for which we have provided our main results.
>
> Besides confirming that our TPSR framework is indeed model-agnostic and can be used as a planning module for future models, we have shown in our manuscript that the current TPSR model applied on E2E performs SOTA in some benchmarks and is a competitor in others. In line with your suggestion, we also investigated a comparison with the mentioned SOTA works [30] and [31]. We would like to note that the code for [31] was released two months after the NeurIPS submission deadline. Additionally, while the code of [30] was partially released approximately a month before our submission (as part of the DSO package), we observed that at least two out of the five main components of this model (pretrained weights and AI-Feynman), along with some details, were not included in their current release. We tried our best to conduct new experiments to compare our results with [30], utilizing their current code version. Results (shown in Table 2 of the attached PDF (please check global response)) represent that TPSR outperforms [30] in black-box datasets, and performs competitively in the Feynman dataset. However, we would like to note that since the current results of [30] are evaluated without some of the main components of the model, we think it is not a fair comparison to be included in the main results of our paper. Also, we again would like to emphasize that being model-agnostic is a much more important aspect of this work as the SR models are rapidly improving.

---

> ### Author Response · Authors · 2023-08-14
> **Looking forward to discussion**
>
> Dear Reviewer zZmW,
>
> Thank you for your feedback during the review process! If there are any concerns or questions, please do not hesitate to let us know - before the author discussion period ends. We will be happy to answer them during the discussion.
>
>
> Thank you,
>
> Paper13018 Authors

---

### Official Review · Reviewer_i1kS · 2023-07-06

**Soundness:** 2 fair
**Presentation:** 2 fair
**Contribution:** 2 fair
**Rating:** 5
**Confidence:** 4

**Summary:**

This paper proposed to combine pretrained Symbolic Regression models with MCTS procedure to improve SR performance without finetuning the pretrained models. Experiments are conducted to demonstrate the improved performance the proposal.

**Strengths:**

1. Proposed a new MCTS based decoding procedure to improve pretrained SR models performance without finetuning the models.
2. The paper is clearly written and easy to follow in texts.
The experiments clearly demonstrates the performance improving over the baselines and the E2E models' decoding.

**Weaknesses:**

1. A few key building blocks needs to be summarized from the literature to be a self-contained paper, e.g. how the datasets are embedded.
2. The methodology contribution is minor as  only MCTS is introduced on top of SR models although experimental performance improvement is observed. Although this is a valuable contribution it might not meet the bar for NEURIPS.

**Questions:**

1. Please provide more statistics of equation (1)'s terms. For example, would N(s) be mostly 1s in your setting? Please be more specific about what "visit count" means exactly. Will the sub-routine beam search generated sequence be accounted as a visit? Will cache hit be counted as visit during beam search sub-routine?
2. Figure 7,  please clarify how you count the number of generated candidates. Are the sub-routine beam search generated equations are counted? Will this difference render the results differently?


Minors:
	1. In the abstract, "GP-based methods" should use the full name of GP for broader readers' convenience.

	2. Line 48, there should be an indent space at the beginning of the sentence.

	3. In section 3, to be self-contained, please succinctly re-iterate the key components of the underlying SR pretrained models, e.g. dataset embedding.

	4. Line 188, please briefly explain why this is still a Q function in standard MDP framework.

	5. Section 4, please be specific whether E2E and TPSR are using exactly the same experiment settings except the MCTS and beam search difference. If possible, a table in the supplemental materials comparing the experiment settings across different approaches might be helpful.

	6. Line 244, in the equation of R^2, is \bar{y} the average values of y in N_{test}? Please clarify.

---

> ### Author Rebuttal · Authors · 2023-08-09
>
> Thank you for the comments and thoughtful questions. Please find our answers below.
>
> **Summary of Pretrained SR Details.** We adopted the pretrained SR model backbone from [18], using its embedding module to embed data points, and leveraging Transformers encoder and decoder modules for representation and expression generation. Given the potential for large input sequences with tokenized numeric data, and the quadratic complexity of Transformers, [18] introduced a linear embedder module to map inputs to a single embedding space before feeding them to the Transformers encoder. We left out these details because of page limits and relied on references. However, we agree on the importance of clarity. We'll make sure to include a summary of these key points in our updated paper.
>
> **Methodology Contribution.** We hope the clarifications provided below address your concern about contribution.
>
> Integration is Not Simply Concatenation: While we leverage both MCTS and pretrained SR models, our contribution isn't a straightforward 'stacking' of the two. The fusion demands a meticulous redesign, modification, and the introduction of new methods to counter the distinct challenges of such integration. This distinction, we believe, is an innovation in itself and is elaborated upon in Appendix C.2 and C.3, with visual insights provided in Fig. 8 and 9.
>
> Differentiating from Related Works: As highlighted in our related works, other works have used MCTS and LLMs for Planning in NLP. A notable example, [34], merges MCTS with a pre-trained discriminator.
> This discriminator can assess both partial and complete states, streamlining its combination with MCTS to determine a node's (or state's) value. Contrarily, our SR equation generation model only allows evaluations and feedback upon equation completion, complicating and increasing the cost of the planning process. We'll need to generate complete equations using beam search simulations and design caching mechanisms to reduce repetitive generation calls to the pretrained SR model.
>
> We want to again emphasize the following contributions:
>
> * We are the first to combine MCTS as a planning-based decoding module with pretrained SR models for the task of equation generation. TPSR not only achieves significantly better performance than the backbone pretrained model but also holds competitive performance compared to the established baselines.
> * We have designed the interfaces between these two components (pretrained SR model and planning search) effectively and dealt with unique challenges of making the framework computationally more efficient.
> * We have also showcased the versatility of TPSR across different equation-specific goals, from fitting accuracy to complexity. Notably, TPSR allows for optimizing equation learning based on varying objectives without necessitating finetuning of the large pretrained SR models.
>
> **Number of Visits Clarification.** The term $N(s)$ in equation (1) represents the "visit count" of state $s$, indicating the number of times that state $s$ has been encountered in the tree search during decoding. In our experiments, the value of $N(s)$ varies greatly depending on the complexity of the symbolic regression task and the state $s$ at hand. For simpler problems, or at early stages of the tree search, $N(s)$ might indeed be closer to 1, as states would not have been explored as thoroughly. For more complex problems, or deeper into the search, $N(s)$ can increase significantly as the same state may be visited multiple times in search. In our MCTS setting, a "visit" means that a state-action pair $(s,a)$ has been passed through during the tree search, and the corresponding child state $s'$ has been added to the tree. Sequences that are generated as part of the beam search sub-routine of simulations in the evaluation stage of MCTS are not directly considered as visits to the nodes corresponding to these sequences. Instead, they serve the purpose of completing the partial equation to allow for feedback computation. As for cache hits, they are also not counted as visits. The reason is that caching in this context is used to save computation by storing previously computed values, and a cache hit simply means retrieving a stored value rather than performing a new visit.
>
> **Number of Generated Candidates.** It's important to note that the visit count is just used in the selection step of the search to promote exploration (as explained in equation (1) ), while the number of generated equation candidates (shown in Fig. 7 as budget), refers to the total number of complete equation sequences that have been generated by each method, i.e., the sample size in the E2E with sampling decoding baseline, and the number of function calls of beam search sub-routine in TPSR (refer to Alg. 1), excluding instances where cached sequences were identified and utilized through sequence caching.
> Thanks for raising these insightful questions! We will make sure to include clarification on these points in the main paper.
>
> **Response to Minor Comments:**
> We will address each of the points you raised in the updated version. Answering some of the questions:
> * **Line 188:** The term $Q$ function here represents the value associated with each node/state $s$ when action $a$ is taken (leading to node $s'$). If the node is part of the trajectory to the current state's root, this value is updated after the backpropagation step.
> * **Section 4:** We assure that both E2E and TPSR use the same experimental settings, with the only difference being the MCTS and beam search/sampling implementations. Some details of the settings are already included in Lines 251-261. We agree that adding a table would make it more clear. We'll make sure to include it in the appendix of the updated version.
> * **Line 244:** You are correct. The reported $R^2$ is on the test set, therefore, $\bar{y}$ represents the average values of $y$ in $N_{test}$. We will add a brief note to clarify this.

---

> > ### Comment · Reviewer_i1kS · 2023-08-20
> >
> > Thank you for responding to my review. I've updated my scores slightly. Please keep improving the paper.

---

> > > ### Author Response · Authors · 2023-08-21
> > > **Thank you**
> > >
> > > Thank you for reviewing our rebuttal and raising the score. We will make sure to update the manuscript in line with your suggestions.

---

> ### Author Response · Authors · 2023-08-14
> **Looking forward to discussion**
>
> Dear Reviewer i1kS,
>
> Thank you for your feedback during the review process! If there are any concerns or questions, please do not hesitate to let us know - before the author discussion period ends. We will be happy to answer them during the discussion.
>
> Thank you,
>
> Paper13018 Authors

---

### Author Rebuttal · Authors · 2023-08-09

We sincerely thank all the reviewers for dedicating their time and expertise to review our manuscript. Please refer to the attached PDF where we have included the Tables and Figures referenced in the subsequent responses. We hope our clarifications address your concerns and look forward to further discussions.

---

### Decision · Program_Chairs · 2023-09-21

**Decision:**

Accept (poster)

**Comment:**

The paper attempts to improve symbolic regression. In this regards, the authors leverage MCTS in the transformer decoding process to tradeoff the fitting accuracy and equation complexity. All the reviews were mostly on positive side, but with some concerns about novelty of the approach. We thank the authors and reviewers for actively engaging during the discussion phase to improve the paper. Author response resolved many concerns of the reviewers and consequently many reviewers increased their scores. Please update the final version of the paper with the clarifications asked by reviewers, guidance on selection of $\lambda$, and an updated contribution list.